# One-Pot Synthesis and Molecular Modeling Studies of New Bioactive Spiro-Oxindoles Based on Uracil Derivatives as SARS-CoV-2 Inhibitors Targeting RNA Polymerase and Spike Glycoprotein

**DOI:** 10.3390/ph15030376

**Published:** 2022-03-20

**Authors:** Samar A. El-Kalyoubi, Ahmed Ragab, Ola A. Abu Ali, Yousry A. Ammar, Mohamed G. Seadawy, Aya Ahmed, Eman A. Fayed

**Affiliations:** 1Pharmaceutical Organic Chemistry Department, Faculty of Pharmacy (Girls), Al-Azhar University, Cairo 11754, Egypt; s.elkalyoubi@hotmail.com; 2Chemistry Department, Faculty of Science (Boys), Al-Azhar University, Cairo 11884, Egypt; 3Department of Chemistry, College of Science, Taif University, P.O. Box 11099, Taif 21944, Saudi Arabia; o.abuali@tu.edu.sa; 4Main Chemical Laboratories, Egypt Army, Cairo 11351, Egypt; biologist202054@yahoo.com; 5Molecular Virology and Immunology Unit, Cancer Institute, Cairo University, Kasr Al-Ainist, El-Khaleeg, Cairo 11976, Egypt; ayaah2223@yahoo.com

**Keywords:** spiro-oxindoles, isatin sulfonamide derivatives, SARS-CoV-2 inhibitory agents, RNA-polymerase (RdRp) and spike glycoprotein inhibition, computational studies

## Abstract

The first outbreak in Wuhan, China, in December 2019 was reported about severe acute coronaviral syndrome 2 (SARS-CoV-2). The global coronavirus disease 2019 (COVID-19) pandemic in 2020 resulted in an extremely high potential for dissemination. No drugs are validated in large-scale studies for significant effectiveness in the clinical treatment of COVID-19 patients, despite the worsening trends of COVID-19. This study aims to design a simple and efficient cyclo-condensation reaction of 6-aminouracil derivatives **2a**–**e** and isatin derivatives **1a**–**c** to synthesize spiro-oxindoles **3a**–**d**, **4a**–**e**, and **5a**–**e**. All compounds were tested in vitro against the SARS-CoV-2. Four spiro[indoline-3,5′-pyrido[2,3-*d*:6,5-*d*’]dipyrimidine derivatives **3a**, **4b**, **4d**, and **4e** showed high activities against the SARS-CoV-2 in plaque reduction assay and were subjected to further RNA-dependent-RNA-polymerase (RdRp) and spike glycoprotein inhibition assay investigations. The four compounds exhibited potent inhibitory activity ranging from 40.23 ± 0.09 to 44.90 ± 0.08 nM and 40.27 ± 0.17 to 44.83 ± 0.16 nM, respectively, when compared with chloroquine as a reference standard, which showed 45 ± 0.02 and 45 ± 0.06 nM against RdRp and spike glycoprotein, respectively. The computational study involving the docking studies of the binding mode inside two proteins ((RdRp) (PDB: 6m71), and (SGp) (PDB: 6VXX)) and geometrical optimization used to generate some molecular parameters were performed for the most active hybrids.

## 1. Introduction

Coronaviruses (CoV) are a type of virus that can cause mild to severe respiratory distress symptoms of cough, high fever, headache, rigor, myalgia, and dizziness [1,2]. Two coronavirus outbreaks, Serious Acute Respiratory Syndrome (SARS) and Middle East Respiratory Syndrome (MERS) have emerged as epidemics with high mortality in the last two decades. SARS CoV transmission from civet cats to humans occurred in China in 2002, and MERS CoV transmission from dromedary camels to humans occurred in Saudi Arabia in 2012 [3]. A cluster of pneumonia cases with an unknown cause surfaced in Wuhan, Hubei Province, China, on 31 December 2019, and China later reported that the outbreak was linked to a seafood market in Wuhan. China revealed the genomic code of the new coronavirus that caused the outbreak for diagnostic purposes on 12 January 2020 [4]. Among other coronaviruses, the virus causing COVID-19 has the advantage of the presence of a unique polybasic cleavage site leading to its increased pathogenicity [5]. In addition, it consists of some structural and some non-structural proteins [6]. The structural proteins include spike (S protein), envelope (E protein), membrane (M protein), nucleocapsid (N protein). In contrast, the non-structural proteins include: main protease (M^pro^), papain-like protease (PL^pro^), non-structural protein 13 (nsp13, helicase), non-structural protein 12 (nsp12, RNA-dependent RNA polymerase), *N*-terminal exoribonuclease and C-terminal guanine-N7 methyl transferase (nsp14), uridylate-specific endoribonuclease (nsp15), 2′ -O-methyltransferase (nsp16) and nsp10 [7,8]. Because heterocyclic chemicals have been implicated in a variety of diseases, including viral infections, AIDS, and cancer, there is a significant potential for using these numerous nuclei to combat coronaviruses. Some antiviral drugs containing indolin-2-one, such as Arbidol and Marboran/Methisazone, are tested against COVID-19 infection. These hybrids, Arbidol **I** and Marboran/Methisazone **II** inhibit membrane fusion and mRNA and protein synthesis, respectively [9]. The use of variants of already recognized antiviral medicines is a useful option until more accurate treatment methodologies for COVID-19 become available [10]. In addition, the pyridine containing hits as 5-chloropyridin-3-yl-1*H*-indole-4-carboxylate (**III**) and chloroquine **IV**, show specificity against SARS CoV 3CL^pro^. All screened compounds showed potential electrophilic centers which may be capable of forming a covalent bond with the nucleophilic thiol of Cys 145 at the active site of 3CL^pro^ [11] (Figure 1).

Furthermore, the purine and pyrimidine derivatives, such as Acyclovir **V** and Ganciclovir **VI** and Lopinavir **VII,** were tested against COVID-19 infection and were shown to be effective in the treatment of COVID-19 [12] involving more than two hydrogen bonds with M^pro^, which were further analyzed by SARS-CoV M^pro^ inhibition assay. Additionally, purine derivatives have been shown to have significant antiviral action against a variety of viruses. Nucleoside analogs, based on purine moiety, were designed and analyzed for their activity against SARS-CoV using plaque reduction assay in SARS CoV Frankfurt-1 strain infected Vero E6 cells [13] (Figure 2).

Molecular hybridization/conjugation is a promising strategy involving combining two pharmacophoric groups via a covalent bond to create a single entity [14]. It was hypothesized that the hybridization of two or more pharmacologically active components in the molecular architecture of hybrid compound/conjugate would prove to be a promising chemotherapeutic agent [15,16,17,18]. As a result of these findings, and as part of our medicinal program aimed at the discovery of novel biologically important heterocyclic compounds with various biological activities [19,20], we integrated the structural features of isatin to design and synthesize a new class of isatin–pyridine-pyrimidine conjugates, hoping to identify novel functional molecules with potent antiviral effects. The results may provide useful information for the design of novel chemotherapeutic drugs (Figure 3).

## 2. Results and Discussion

### 2.1. Chemistry

The synthetic strategy to design the target spiro-oxindoles based on uracil derivatives are outlined from Figure 1, Figure 2 and Figure 3. Both 5-(substituted-1-ylsulfonyl)indoline-2,3-dione **1a**–**c** and 6-aminouracil **2a**–**e** were chosen as the scaffold for annulations of the target congeners in our quest to create new spiro-oxindoles based on uracil derivatives with potential pharmacological significance via one-pot reaction.

The substituted 6-aminouracil derivatives **2a**–**e** were obtained by adding urea, methyl urea, and/or methyl thiourea to ethyl cyanoacetate in absolute ethanol containing sodium (sodium ethoxide) by heating under reflux, according to the reported methods [21,22]. Additionally, 5-(substituted-sulfonyl)indoline-2,3-dione **1a**–**c** were prepared according to the previously reported method [23,24].

By heating 6-aminouracil derivatives **2a**–**e** under reflux condition with 5-(substituted-sulfonyl)indoline-2,3-dione **1a**–**c**, a series of **3a**–**d**, **4a**–**e**, and **5a**–**e** in a moderate yield were obtained (Figure 1 and Figure 3). The mechanistic pathway for the synthesized 1′*H*-spiro[indoline-3,5′-pyrido[2,3-*d*:6,5-*d*′]dipyrimidine derivatives **3a**–**d**, **4a**–**e**, and **5a**–**e** were illustrated in Figure 2. The first step involved condensation of C3-carbonyl functionality in indoline-2,3-dione derivatives with C5-in 6-aminouracil derivatives in the presence of acetic acid to form 5-(2-oxo-5-(substituted-sulfonyl)indolin-3-ylidene)dihydropyrimidine derivatives **B**. The arylidene intermediate **B** reacted further with a second mole of 6-aminouracil derivatives to obtain 2-oxoindolin-3-yl derivatives **C** that underwent cyclization through nucleophilic addition followed by elimination of ammonia molecule to get the desired product. The IR spectra of compound **3a** showed absorption bands at *ʋ* 3330, 3187, 1702, and 1662 cm^−1^ corresponding to NH, and carbonyl groups. Its ^1^H NMR spectra displayed four exchangeable singlet signals at *δ* 9.72, 10.87, 11.06, 11.71 ppm, and three signals for three aromatic protons of indolin-2-one derivative as two doublets at *δ* 7.16, 7.51 ppm and one doublet of a doublet at *δ* 7.65 ppm. Moreover, two singlet signals at *δ* 3.34, 3.45 ppm for two methyl protons (2CH_3_) and eight pyrrolidinyl (2CH_2_) appear as two multiplet signals at *δ* 1.55, 3.02 ppm.

Additionally, the ^1^H NMR spectra of 1′,9′-*bis*(4-chlorobenzyl)-1′*H*-spiro[indoline-3,5′-pyrido[2,3-*d*:6,5-*d*′]dipyrimidine **4d** revealed four exchangeable signals at *δ* 9.84, 11.15, 11.38, 11.90 ppm attributed to four NH groups. Moreover, signals at *δ* 5.18, 5.28 ppm owned to four protons of two methylene groups (2CH_2_) and signals at *δ* 1.32, 1.45, 2.81 ppm related to piperidinyl protons (5CH_2_). In addition, the eleven aromatic protons ranged between *δ* 7.06–7.59 ppm. Furthermore, the ^13^C NMR spectra of compound **5d** exhibited four signals at *δ* 21.06, 22.84, 24.46, 46.34 ppm related to piperidinyl carbons, and two methylene carbons appeared at *δ* 44.61 ppm and spiro-carbon at *δ* 50.30 ppm. Furthermore, the aromatic carbons ranged between 83.90–146.75 ppm, as well as signals at 149.58, 150.15, 153.29, 157.84, 160.75, 172.03, 180.94 ppm corresponding to ethylenic carbon attached to nitrogen (2C=C-N) and five carbonyl groups. (Appendix A involving the IR, ^1^H and ^13^C NMR data of all the synthetized compounds were represented in the Appendix A).

### 2.2. Antiviral Activity

#### 2.2.1. The Half-Maximal Inhibitory Concentration (IC_50_)

Eventually, the development of an effective antiviral for SARS-CoV-2, if given to patients early in infection, could help to limit the viral load, prevent severe disease progression, and limit person–person transmission. Benchmark testing of those compounds against other potential antivirals for SARS-CoV-2 with alternative mechanisms of action would therefore be important as soon as practicable [25]. The synthetic compounds **3a**–**d**, **4a**–**e**, and **5a**–**e** were tested in vitro against the SARS-CoV-2 strain isolated from Egyptian patients. The reference drug was chloroquine, a well-known chemotherapeutic agent. The results were presented as IC_50_ values and described in the table below (Table 1 and Figure 4). The effect of different concentrations of the compounds on the cellular proliferation of the Vero E6 cell line following 24 h of treatment was determined using MTT assay [26]. According to the results, most of the compounds tested showed moderate to excellent cytotoxic activity against SARS-CoV-2, ranging from 4.10–5.93 µM compared with chloroquine as the standard drug with IC_50_ value of about 2.24 µM.

#### 2.2.2. Plaque Reduction Assay (% of Inhibition SARS-CoV2)

As represented in Table 2 and Figure 5, four spiro[indoline-3,5′-pyrido[2,3-*d*:6,5-*d′*]dipyrimidine derivatives **3a**, **4b**, **4d**, and **4e** showed high activities against the SARS-CoV-2 that was isolated from Egyptian patients. Firstly, 5-(pyrrolidin-1-ylsulfonyl)-1′*H*-spiroindoline derivative **3a**, which has R = CH_3_ and X = O, found the most active derivatives among these series **3a**–**d** against replication of the virus with the percentage of inhibition = 84%. Additionally, replacing the methyl group with a benzyl or 4-Cl-benzyl moiety, as in compounds **3b** and **3c**, the activity decreased to 75% and 0%, respectively. This decrease in the activity because of the presence of benzylidene moiety in general, and the substitution of one hydrogen bond by chlorine atom at position four in the benzylidene moiety, leads to removing the activity.

On the other hand, substituting the carbonyl by thiocarbonyl (X = S), as well as presence of the methyl group as described in 5-(pyrrolidin-1-ylsulfonyl)-2′,8′-dithioxo-spiroindoline derivative **3d**, displayed the percentage of inhibition as 55%. Furthermore, 5-(piperidin-1-ylsulfonyl)-1′*H*-spiroindoline derivatives **4b** and **4e** with a methyl moiety and carbonyl or thiocarbonyl (X = O or S) showed the highest activity among the tested derivatives against replication of SARS-CoV-2 with a percentage of inhibition reading 99% and 91%, respectively. Moreover, 1′*H*-spiroindoline derivative **4c** demonstrated an inhibition percentage = 74%, and this decrease in activity may be related to the presence of benzyl moiety. Additionally, compound **4d** exhibited good activity with the percentage of inhibition 80%, when the substitutions were R = 4-chloro benzyl and X = O.

Finally, 5-(morpholinosulfonyl)-1′*H*-spiroindoline derivatives **5a**–**e** revealed weak antiviral activity against SARS-CoV-2, with a percentage of inhibition ranging from 20–70%. Furthermore, in pharmacological terms, the presence of the pyrrolidin-1-yl and piperidin-1-yl moieties on the skeleton of 1′*H*-spiroindoline derivatives **3a**–**d** and **4a**–**e** may be directly responsible for their antiviral activities. In contrast, the presence of the morpholino moiety as in the series **5a**–**e** exhibited weak antiviral agents.

#### 2.2.3. In Vitro Enzymes Assay

The RNA-dependent RNA-polymerase (RdRp) and spike glycoprotein inhibition assay were investigated for the most potent anti-SARS-CoV-2 active hybrids **3a**, **4b**, **4d**, and **4e** using chloroquine as a reference drug. The assessment results were summarized in Table 3 as IC_50_ values in nM. All tested compounds exhibited potent inhibitory activity towards RdRp and spike glycoprotein ranging from 40.23 ± 0.09 to 44.90 ± 0.08 nM and 40.27 ± 0.17 to 44.83 ± 0.16 nM, respectively, when compared with chloroquine as a reference standard, which showed 45 ± 0.02 and 45 ± 0.06 nM against RdRp and spike glycoprotein, respectively.

It was found that 5-(pyrrolidin-1-ylsulfonyl)-1′*H*-spiroindoline derivative **3a** strongly inhibited RdRp and spike glycoprotein (IC_50_ = 40.23 ± 0.09 and 40.27 ± 0.17 nM, respectively) and had relatively higher potency than the standard (IC_50_= 45 ± 0.02 and 45 ± 0.06 nM). This increase of the reactivity of 1′,9′-dimethyl-5-(pyrrolidin-1-ylsulfonyl)-1′*H*-spiro[indoline-3,5′-pyrido[2,3-*d*:6,5-*d*′]dipyrimidine]-2,2′,4′,6′,8′(3′*H*,7′*H*,9′*H*,10′*H*)penta-one (**3a**) may be due to the presence of pyridyl moiety, in addition to *N*-methyl and X = O. Upon replacement of the oxygen atom with sulfur as in **3d**, the activity diminished.

Moreover, 5-(piperidin-1-ylsulfonyl)-1′*H*-spiroindoline derivatives **4d** and **4e** revealed moderate inhibitory activity against RdRp (IC_50_= 41.26 ± 0.25 and 42.27 ± 0.31) and spike glycoprotein (IC_50_= 41.23 ± 0.12 and 42.27 ± 0.31 nM), respectively. On the other hand, compound **4b** revealed equipotent activity to the used reference chloroquine with IC_50_= 44.90 ± 0.08 and 44.83 ± 0.16 nM.

Finally, it can be concluded that the existence of pyrrolidinyl/piperidinyl moiety, in addition to the *N*-substituted with methyl or 4-chlorobenzyl and X = O as 1′*H*-spiroindoline derivatives **4b** and **4d** is essential for antiviral activity (Table 3).

### 2.3. Computational Study

#### 2.3.1. Computational Study of the Binding Mode

Recently, in regards to structure-based drug design, molecular docking has become an evermore essential medication discovery method and is considered the most common approach in pharmaceutical research [27,28]. Additionally, the docking study illustrates the interaction between newly synthesized compounds and active sites in selected proteins at the atomic level to determine the behavior of new promising compounds in the binding site of the target proteins [29]. Furthermore, docking simulation studies have been progressed to visualize, calculate, formulate, and hypothesize about the energy and orientation of new ligands in the active site in the pocket of a target protein [30,31]. Our study involved docking of the most active derivative inside the RNA polymerase (RdRp) (PDB: 6m71) and spike glycoprotein (SGp) (PDB: 6VXX), and the results displayed good binding energy with lower binding affinity values (S) (kcal/mol) with different types of interactions and the docking results represented in Table 4 and Figure 6, Figure 7, Figure 8 and Figure 9.


**Docking Study Inside RNA-Dependent RNA Polymerase (RdRp) (PDB: 6m71)**


Docking of the most active derivatives **3a**, **4b**, **4d**, and **4e,** inside the active site of RNA polymerase (RdRp) (PDB: 6m71), exhibited lower binding energy ranging from −18.48 kcal/mol to −15.38 kcal/mol compared with chloroquine as positive control S = −14.94 kcal/mol and remdesivir S = −16.09 Kcal/mol. All the designed derivatives displayed lower binding energy than chloroquine with different types of interaction (H-bonds donor or H-bond acceptors and arene–cation interaction besides hydrophobic interactions). Furthermore, the four derivatives displayed hydrophobic interaction through an aliphatic group (pyrrolidinyl, piperidinyl, and morphinyl), oxygen of sulfonyl, phenyl of indolin-2-one, alongside carbonyl and methyl, and NH of pyridino-pyrimidine moiety with the adjacent residues.

Additionally, compound **3a** revealed two hydrogen bonds donor between the amino acid residues, Asp623 and Lys621, with the carbonyl of indolin-2-one and the carbonyl of the pyridino-pyrimidine derivative with bond lengths of 2.90 and 2.70 Å, respectively. In addition, there were two hydrogen bond acceptors between the oxygen of sulfonyl derivatives with Arg553 and Arg555 with bond lengths of 2.83 Å and 2.74 Å (Figure 6), respectively.

Furthermore, compound **4b** displayed two hydrogen bond acceptors between Lys621 and carbonyl of indolin-2-one derivative, Arg553 and oxygen of sulfonyl, and one hydrogen bond donor between Asp623 with NH of pyridino-pyrimidine moiety with bond lengths of 2.88 Å (30%), 2.73 Å (32%) and 2.56 Å (70%), respectively. Moreover, compound **4d** that excreted the highest binding energy S = −18.48 Kcal/mol, showed only one hydrogen bond side-chain acceptor between Lys551 and carbonyl of a pyridino-pyrimidine nucleus with 2.63 Å bond length and 44% strength. In addition, compound **4d** can form two arene–cation interactions between the two-aryl group of benzylidene derivatives with Lys708 and Arg553, alongside hydrophobic interactions (Figure 7).

On the other hand, compound **4e** exhibited the lowest binding energy of the synthesized derivatives. It also exhibited less binding energy than chloroquine (with the binding energy S = −15.38 Kcal/mol) with three hydrogen bonds between Asp760 and Asp623 with NH of a pyridino-pyrimidine nucleus, and Arg553 with the oxygen of sulfonyl derivatives with bond lengths and strengths of 2.38 Å (48%), 2.81 Å (16%), and 2.53 Å (71%), respectively. (All docking figures are represented in the Appendix A).

Finally, chloroquine established binding energy–14.94 Kcal/mol through a one side-chain hydrogen bond donor between the Asp760 and NH of quaternary salt in the side chain with a bond length of 2.42 Å. In addition, the arene–cation interaction between Arg553 and phenyl ring of quinoline ring occurred. Additionally, the remdesivir demonstrated binding energy S = −16.09 Kcal/mol, through three hydrogen bond side-chain acceptors with the residues Arg553, Arg553, alongside one hydrogen bond side-chain donor with Asp623 and one hydrogen bond backbone donor with Thr556 (See Table 4 and the Appendix A for more details).


**Docking Study Inside Spike Glycoprotein (SGp) (PDB: 6VXX)**


The simulation study was extended to study the reactivity and interaction of the newly designed derivatives inside the active site of spike glycoprotein as a second target for these derivatives. The docking score energy showed lower binding energy S = (−17.67 to −15.22 Kcal/mol) in comparison with chloroquine S = −15.71 Kcal/mol and remdesivir S = −15.67 Kcal/mol. Docking simulation results displayed that these derivatives were chiefly combined in the form of hydrogen bonds with good binding affinity indicating that the synthesized derivatives could hinder the binding of the SARS-CoV-2 spike glycoprotein domain.

Compound **3a** demonstrated binding energy S = −17.67 Kcal/mol, through two side-chain hydrogen bond acceptors between His1058, Thr732 and carbonyl of the pyridino-pyrimidine with bond lengths of 2.64 Å (35%) and 2.66 Å (77%), and one hydrogen bond side-chain donor between Thr732 with NH of a pyridino-pyrimidine derivatives with a bond length of 2.73 Å (Figure 8).

Moreover, compounds **4b** and **4e** showed binding energy S = −15.22, −15.92 Kcal/mol, respectively, through two hydrogen bonds with the bond length ranging between 2.39–2.72 Å. Noticeably, compound **4d** revealed binding energy S = −16.48 Kcal/mol with one hydrogen bond backbone donor between Leu861 and NH of pyridino-pyrimidine derivative with a bond length of 2.83 Å (Figure 9). Last of all, chloroquine as positive control advertised binding energy S = −15.71 Kcal/mol with two hydrogen bond side-chain donors between Asp867 and NH of chloroquine at C4 and NH of quaternary salt at the side chain with a bond length of 2.71, 2.48 Å, alongside arene–cation interaction between His1058 and nitrogen of quaternary salts as well as hydrophobic interaction. Additionally, the remdesivir revealed binding energy S = −15.67 Kcal/mol through only one backbone donor between the residue Phe823 and amino group of pyrrolo[2,1-*f*] [1,2,4]triazin derivative with a 3.01 Å bond length and 22% strength (See all docking figures in the Appendix A).

Finally, it can be concluded that the combination of the indolin-2-one sulfonamide derivatives with 6-aminouracil derivatives to form a new hybrid target as 5-(substitutedsulfonyl)-1*′H*-spiro[indoline-3,5′-pyrido[2,3-*d*:6,5-*d*′]dipyrimidine derivatives will probably hinder the binding of SARS-CoV-2 RNA polymerase and spike glycoprotein more effectively. The docking results were confirmed by the IC_50_ and inhibition percentage values in experimental activity and it can be said that these derivatives could resist the SARS-CoV-2.

#### 2.3.2. Geometrical Optimization and Molecular Parameters

The molecular modelling calculation using the DFT method was carried out to evaluate and specify the relationship between the structure of the most active derivatives and the experimental activity results. Additionally, the geometrical optimization of the most promising derivatives **3a**, **4b**, **4d**, **4e** and positive controls (**Chloroquine** and **Remdesivir**) were calculated by the DFT method that was used to determine the frontier energies of molecular orbital, and therefore deduce some quantum molecular parameters according to previously reported methods [32,33], as described in Table 5 and Table 6.

The results showed the highest occupied molecular orbitals E_HOMO_ ranged between (−6.17 to −6.00 eV) and the lowest unoccupied molecular orbitals E_LUMO_ between (−2.13 to −1.74 eV) compared with chloroquine (E_HOMO_ = −5.66 eV and E_LUMO_ = −1.17 eV) and remdesivir (E_HOMO_ = −6.11 eV and E_LUMO_ = −1.27 eV). The HOMO energy represented the ability of these derivatives to give electrons as electron donors and localized mainly on 2-oxoindoline derivatives. On the other hand, the LUMO energy displayed by a site that has the ability to act as an electron attractive, i.e., electron acceptors due to vacant orbitals and localized at pyrido[2,3-*d*:6,5-*d*′]dipyrimidine derivatives moiety.

Furthermore, the most active derivatives **3a**, **4b**, **4d**, and **4e** revealed a lower energy gap (ΔE) (4.03–4.26 eV) compared with chloroquine (ΔE) = 4.49 eV, and remdesivir (ΔE) = 4.83 eV, i.e., the synthesized derivatives displayed the most stable conformer as well as the most polarizable. Generally, the molecules with lower energy gaps required less excitation energy and could offer electrons to neighboring biological receptors and were predicted to have high biological potency as confirmed by the experimental results [34]. Moreover, the values of softness *S* = 0.469–0.495 eV^−1^ and hardness *ɳ* = 2.01–2.13 eV displayed superior activity and lower hardness for the most active derivatives **3a**, **4b**, **4d**, and **4e** in comparison to chloroquine (*S* = 0.446 eV^−1^ and *ɳ* = 2.24 eV) and remdesivir (*S* = 0.41 eV^−1^ and *ɳ* = 2.41 eV). Additionally, all the tested derivatives exhibited higher electrophilic index (*ω*) = 3.52–4.27 eV rather than chloroquine (*ω*) = 2.59 eV and remdesivir (*ω*) = 2.81 eV.

#### 2.3.3. Molecular Electrostatic Potential (MEP)

Designing new drugs via molecular electrostatic potential (MEP) is an important property in evaluating the shape, size, and charge distribution around the molecules. The distribution of electron density on the surface of molecules provides us information with the regions that have the ability to donate electrons (which act as nucleophiles) and accept electrons (which act as electrophiles) by appearing in different colors. The electrostatic potential increase is in order of blue > green > yellow > orange > red. The red region indicated an electron-rich region (negative charge sites), while the blue region designated an electron deficiency region (partially positive charged), and the green and yellow regions indicated the neutral sites [35,36].

The MEP map of the most promising derivatives **3a**, **4b**, **4d**, **4e** and positive controls (**Chloroquine** and **Remdesivir**) are presented in Figure 4. The electron-rich areas are represented at the oxygen of indolin-2-one derivatives, oxygen of sulfonyl group, and oxygen of carbonyl of pyrido[2,3-*d*:6,5-*d*′]dipyrimidine derivatives moiety, as well as nitrogen of quinoline in chloroquine, while an electron deficiency region is located at the nitrogen of pyrido[2,3-*d*:6,5-*d*′]dipyrimidine in spiro derivatives. Moreover, the yellow and green colors appear on carbon and hydrogen of all designed and chloroquine derivatives that characterized neutral sites as represented in Figure 10. The MEP of the most promising derivatives **3a**, **4b**, **4d**, and **4e** are abundant with positive and negative regions that are important in the interaction with biological targets, and these results supported a molecular docking study where the hydrogen bonds formed (donors or acceptors) between these regions are different on the active site than in the pocket, as represented in Table 4.

## 3. Materials and Methods

### 3.1. Chemistry

All melting points were measured with the Electrothermal LA9000 sequence, and the Digital Melting Point Apparatus was left uncorrected. At the Pharmaceutical Analytical Unit, Faculty of Pharmacy, Al-Azhar University, the IR Spectra were calculated using the KBr disc technique on a Nikolet IR 200 FT IR Spectrophotometer, and the values are expressed in (cm^−1^). The ^1^H NMR and ^13^C NMR Spectra were recorded on a Bruker 400 MHz Spectrometer and the ^13^C-NMR spectra was ran at 125 MHz in dimethyl sulfoxide (DMSO-*d*_6_) at Applied Nucleic Acid Research Center, Zagazig University, Egypt. Chemical changes were calculated in ppm relative to TMS as an internal norm using DMSO-*d*_6_ as a solvent. At the Regional Center for Mycology and Biotechnology (RCMB) at Al-Azhar University, the mass spectrum was recorded at 70 ev on the DI-50 unit of the Schimadzu GC/ MS- QP5050A Spectrometer and represented as **m*/*z** (relative abundance percent). Moreover, the elemental analysis (C, H, N) was performed at Al-Azhar University’s Regional Center for Mycology and Biotechnology, and the results were found to be within 0.4 percent of theoretical values, unless otherwise mentioned. The biological activities were performed at the Ministry of Defense’s Chemical Warfare Department’s Main Chemical Warfare Laboratories. TLC sheets precoated with UV fluorescent silica gel Merck 60 F254 plates were used to track the progress of the reaction, which was visualized using the UV lamp.


**Synthesis of 1*H*-spiro[indoline-3,5′-pyrido[2,3-*d*:6,5-*d′*]dipyrimidine derivatives**


A mixture of isatin derivatives **1a**–**c** (0.01 mol) and 6-aminouracil derivatives **2a**–**e** (0.02 mol) in acetic acid (7 mL) were heated under reflux for 6–8 hs. The product therefore formed was collected, filtrated and recrystallized from the proper solvent.


**1′,9′-Dimethyl-5-(pyrrolidin-1-ylsulfonyl)-1′*H*-spiro[indoline-3,5′-pyrido[2,3-*d*:6,5-*d*′]dipy-rimidine]-2,2′,4′,6′,8′(3′*H*,7′*H*,9′*H*,10′*H*)-pentaone (3a)**


Yield 72% as Off white powder; M.P.: 350–352 °C; IR: ν/cm^−1^: 3330, 3187 (NH), 3050 (CH-Ar), 2950, 2860 (CH.Aliph.), 1702, 1662 (C=O); ^1^H NMR (400 MHz, DMSO) *δ*/ppm 11.71, 11.06, 10.87, 9.72 (4s, 4H, 4NH exchangeable by D_2_O), 7.65 (dd, *J* = 8.4, 2.1 Hz, 1H, Ar-H), 7.51 (d, *J* = 8.4 Hz, 1H, Ar-H), 7.16 (d, 1H, Ar-H), 3.45, 3.34 (2s, 6H, 2CH_3_), 3.02 (m, 4H), 1.55 (m, 4H); ^13^C NMR (101 MHz, DMSO) *δ* 181.11, 172.03, 160.49, 157.64, 150.69 (5C=O), 149.91, 146.78 (2C=C-N), 139.68, 130.58, 127.49, 125.06, 121.68, 117.60, 97.95, 83.76, 50.23 (C-spiro), 47.86 (2CH_2_-pyrolidine), 30.34, 29.27 (2CH_3_), 24.56 (2CH_2_-pyrolidine); MS (EI, 70 eV): *m*/*z* (%) = 527 (M^+^) (24.58%), 202 (100%); Anal. Calcd for C_22_H_21_N_7_O_7_S (527.51): C, 50.09; H, 4.01; N, 18.59; Found C, 50.21; H, 3.87; N, 18.48%.


**1′,9′-Dibenzyl-5-(pyrrolidin-1-ylsulfonyl)-1′*H*-spiro[indoline-3,5′-pyrido[2,3-*d*:6,5-*d′*]dipy-rimidine]-2,2′,4′,6′,8′(3′*H*,7′*H*,9′*H*,10′*H*)-pentaone (3b)**


Yield 75% as light rose powder; M.P.: 346–348 °C; IR: ν/cm^−1^: 3289, 3250 (NH), 3060 (CH-Ar), 2974, 2850 (CH-Aliph.), 1715, 1644 (C=O); ^1^H NMR (400 MHz, DMSO) *δ*/ppm 11.91, 11.25, 11.04, 9.79 (4s, 4H, 4NH exchangeable by D_2_O), 7.62 (dd, *J* = 8.4, 2.0 Hz, 1H, Ar-H), 7.40 (t, *J* = 7.2 Hz, 2H, Ar-H), 7.35 (d, *J* = 7.2 Hz, 1H, Ar-H), 7.31 (d, *J* = 7.2 Hz, 3H, Ar-H), 7.27 (d, 2H, Ar-H), 7.23 (t, *J* = 6.2 Hz, 3H, Ar-H), 7.15 (d, 1H, Ar-H), 5.01 (d, 2H, CH_2_), 4.90 (d, 2H, CH_2_), 2.95 (t, *J* = 6.6 Hz, 4H, 2CH_2_-pyrolidine), 1.52 (t, *J* = 6.6 Hz, 4H, 2CH_2_-pyrolidine); ^13^C NMR (101 MHz, DMSO) *δ* 181.00, 172.22, 160.95, 157.64, 150.26 (5C=O), 144.79 (2C=C-N), 139.26, 137.51, 135.98, 130.16, 128.79, 128.28, 127.58, 127.44, 126.18, 125.13, 121.41, 117.28, 98.13, 82.57, 50.25 (C-spiro), 47.82 (2CH_2_-pyrolidine), 45.94 (2CH_2_), 24.52 (2CH_2_-pyrolidine); MS (EI, 70 eV): *m*/*z* (%) = 679 (M^+^) (24.58%), 317 (100%); Anal. Calcd for C_34_H_29_N_7_O_7_S (679.71): C, 60.08; H, 4.30; N, 14.43; Found C, 60.35; H, 4.11; N, 14.65%.


**1′,9′-Bis(4-chlorobenzyl)-5-(pyrrolidin-1-ylsulfonyl)-1′*H*-spiro[indoline-3,5′-pyrido[2,3-*d*:-6,5-*d′*]dipyrimidine]-2,2′,4′,6′,8′(3′*H*,7′*H*,9′*H*,10′*H*)-pentaone (3c)**


Yield 78% as rose powder; M.P.: 300–302 °C; IR: ν/cm^−1^: 3320 (NH), 3060 (CH-Ar), 2920, 2855 (CH.Aliph.), 1770, 1727 (C=O); ^1^H NMR (400 MHz, DMSO) *δ*/ppm 11.82, 11.38, 11.13, 9.91 (4s, 4H, 4NH exchangeable by D_2_O), 7.67 (d, *J* = 8.4 Hz, 1H, Ar-H), 7.64 (d, *J* = 7.2 Hz, 1H, Ar-H), 7.56 (d, *J* = 8.0 Hz, 1H, Ar-H), 7.52 (d, *J* = 8.0 Hz, 1H, Ar-H), 7.47 (dd, *J* = 7.6, 1.8 Hz, 1H, Ar-H), 7.36 (d, *J* = 8.0 Hz, 1H, Ar-H), 7.27 (d, 1H, Ar-H), 7.23 (d, *J* = 7.6 Hz, 1H, Ar-H), 7.06 (d, *J* = 6.8 Hz, 1H, Ar-H), 6.98 (d, *J* = 7.2 Hz, 1H, Ar-H), 6.83 (d, *J* = 6.8 Hz, 1H, Ar-H), 5.11 (d, 2H, CH_2_), 4.99 (d, 2H, CH_2_), 3.03 (s, 4H, 2CH_2_-pyrolidine), 1.58–1.55 (m, 4H, 2CH_2_-pyrolidine); ^13^C NMR (101 MHz, DMSO) *δ* 180.8, 166.13, 160.75, 157.66, 150.12 (5C=O), 145.15 (2C=C-N), 139.31, 134.32, 133.32, 131.55, 130.22, 129.58, 129.31, 128.52, 127.19, 126.19, 125.56, 98.41, 83.55, 50.31 (C-spiro), 47.85 (2CH_2_-pyrolidine), 44.50 (2CH_2_), 24.55 (2CH_2_-pyrolidine); MS (EI, 70 eV): *m*/*z* (%) = 747 (M^+^) (25.48%), 749 (M^+2^) (13.44%), 751 (M^+4^) (21.96), 339 (100%); Anal. Calcd for C_34_H_27_Cl_2_N_7_O_7_S (748.59): C, 54.55; H, 3.64; N, 13.10; Found C, 54.84; H, 3.45; N, 13.19%.


**1′,9′-Dimethyl-5-(pyrrolidin-1-ylsulfonyl)-2′,8′-dithioxo-2′,3′,8′,9′-tetrahydro-1′*H*-spiro-[indoline-3,5′-pyrido[2,3-*d*:6,5-*d′*]dipyrimidine]-2,4′,6′(7′*H*,10′*H*)-trione (3d)**


Yield 69% as Off white powder; M.P.: 325–327 °C; IR: ν/cm^−1^: 3253, 3179 (NH), 3106 (CH-Ar), 2930, 2860 (CH-Aliph.), 1769, 1669 (C=O); ^1^H NMR (400 MHz, DMSO) *δ*/ppm 12.55, 12.44, 11.94, 9.93 (4s, 4H, 4NH exchangeable by D_2_O), 7.70 (dd, *J* = 8.6, 2.0 Hz, 1H, Ar-H), 7.57 (d, *J* = 8.6 Hz, 1H, Ar-H), 7.24 (d, 1H, Ar-H), 3.96, 3.74 (2s, 6H, 2CH_3_), 3.09–2.96 (m, 4H, 2CH_2_-pyrolidine), 1.55 (m, *J* = 6.8 Hz, 4H, 2CH_2_-pyrolidine); ^13^C NMR (101 MHz, DMSO) *δ* 180.29, 176.32 (2C=S), 175.56, 157.80, 155.27 (3C=O), 153.71, 147.02 (2C=C-N), 139.50, 131.41, 128.09, 125.16, 120.61, 118.28, 102.23, 88.34, 50.61 (C-spiro), 48.03 (2CH_2_-pyrolidine), 36.81, 36.46 (2CH_3_), 24.74 (2CH_2_-pyrolidine); MS (EI, 70 eV): *m*/*z* (%) = 559 (M^+^) (23.65%), 497 (100%); Anal. Calcd for C_22_H_21_N_7_O_5_S_3_ (559.63): C, 47.22; H, 3.78; N, 17.52; Found C, 47.06; H, 3.88; N, 17.75%.


**5-(Piperidin-1-ylsulfonyl)-1′*H*-spiro[indoline-3,5′-pyrido[2,3-*d*:6,5-*d′*]dipyrimidine]-2,2′,4′,-6′,8′(3′*H*,7′*H*,9′*H*,10′*H*)-pentaone (4a)**


Yield 74% as Yellowish white; M.P.: 360–361 °C; IR: ν/cm^−1^: 3173 (NH), 3065 (CH-Ar), 2953, 2856 (CH-Aliph.), 1693, 1641 (C=O); ^1^H NMR (400 MHz, DMSO) *δ*/ppm 12.00, 11.25 (2s, 2H, 2NH exchangeable by D_2_O), 10.78 (s, 2H, 2NH exchangeable by D_2_O), 10.58, 9.68 (2s, 2H, 2NH exchangeable by D_2_O), 7.51 (dd, *J* = 8.4, 2.0 Hz, 1H, Ar-H), 7.23 (d, *J* = 8.4 Hz, 1H, Ar-H), 7.00 (d, 1H, Ar-H), 2.76 (s, 4H, 2CH_2_-piperidine), 1.44 (s, 4H, 2CH_2_-piperidine), 1.33 (s, 2H, CH_2_-piperidine); ^13^C NMR (101 MHz, DMSO) *δ* 180.78, 165.81, 161.96, 158.63, 153.67 (5C=O), 149.85, 145.83 (2C=C-N), 139.39, 129.67, 127.57, 125.38, 121.79, 116.98, 97.50, 82.79, 49.05 (C-spiro), 46.39, 24.46, 22.92 (5CH_2_-piperidine); MS (EI, 70 eV): *m*/*z* (%) = 513 (M^+^) (15.72%), 186 (100%); Anal. Calcd for C_21_H_19_N_7_O_7_S (513.49): C, 49.12; H, 3.73; N, 19.09; Found C, 49.02; H, 3.85; N, 19.19%.


**1′,9′-Dimethyl-5-(piperidin-1-ylsulfonyl)-1′*H*-spiro[indoline-3,5′-pyrido[2,3-*d*:6,5-*d′*]dipyri-midine]-2,2′,4′,6′,8′(3′*H*,7′*H*,9′*H*,10′*H*)-pentaone (4b)**


Yield 70.5% as Off white powder; M.P.: 349–350 °C; IR: ν/cm^−1^: 3365, 3172 (NH), 3050 (CH-Ar), 2958, 2857 (CH-Aliph.), 1725, 1670 (C=O); ^1^H NMR (400 MHz, DMSO) *δ*/ppm 11.72, 11.07, 10.87, 9.73 (4s, 4H, 4NH exchangeable by D_2_O), 7.58 (dd, *J* = 8.4, 2.0 Hz, 1H, Ar-H), 7.51 (d, *J* = 8.4 Hz, 1H, Ar-H), 7.10 (d, 1H, Ar-H), 3.45, 3.35 (2s, 6H, 2CH_3_), 2.80 (t, 4H, 2CH_2_-piperidine), 1.43 (s, 4H, 2CH_2_-piperidine), 1.31 (s, 2H, CH_2_-piperidine); ^13^C NMR (101 MHz, DMSO) *δ* 181.05, 172.05, 160.51, 157.66, 150.69 (5C=O), 149.92, 146.80 (2C=C-N), 139.66, 130.33, 127.53, 125.30, 121.62, 117.57, 98.09, 83.79, 50.25 (C-spiro), 46.38 (2CH_2_-piperidine), 30.33, 29.29 (2CH_3_), 24.42, 22.94, 21.08 (3CH_2_-piperidine); MS (EI, 70 eV): *m*/*z* (%) = 541 (M^+^) (22.65%), 47 (100%); Anal. Calcd for C_23_H_23_N_7_O_7_S (541.54): C, 51.01; H, 4.28; N, 18.11; Found C, 51.17; H, 4.48; N, 18.02%.


**1′,9′-Dibenzyl-5-(piperidin-1-ylsulfonyl)-1′*H*-spiro[indoline-3,5′-pyrido[2,3-*d*:6,5-*d′*]dipyri-midine]-2,2′,4′,6′,8′(3′*H*,7′*H*,9′*H*,10′*H*)-pentaone (4c)**


Yield 74% as Pink powder; M.P.: 298–299 °C; IR: ν/cm^−1^: 3390, 3190 (NH), 3070 (CH-Ar), 2945, 2856 (CH-Aliph.), 1708,1640 (C=O); ^1^H NMR (400 MHz, DMSO) *δ*/ppm 11.95, 11.27, 11.03, 10.09 (4s, 4H, 4NH exchangeable by D_2_O), 7.79 (d, *J* = 8.0 Hz, 1H, Ar-H), 7.74 (d, *J* = 8.4 Hz, 1H, Ar-H), 7.66 (d, *J* = 8.0 Hz, 1H, Ar-H), 7.62 (d, *J* = 7.6 Hz, 1H, Ar-H), 7.57 (dd, *J* = 8.0, 2.0 Hz, 1H, Ar-H), 7.46 (d, *J* = 8.4 Hz, 1H, Ar-H), 7.37 (t, 2H, Ar-H), 7.13 (t, 2H, Ar-H), 7.11 (d, *J* = 7.2 Hz, 2H, Ar-H), 7.01 (d, *J* = 7.6 Hz, 1H, Ar-H), 5.22 (d, 2H, CH_2_), 5.01 (d, 2H, CH_2_), 2.70 (s, 4H, 2CH_2_-piperidine), 1.38 (s, 4H, 2CH_2_-piperidine), 1.29 (s, 2H, CH_2_-piperidine); ^13^C NMR (101 MHz, DMSO) *δ* 181.15, 171.87, 160.75, 157.47, 150.35 (5C=O), 149.35, 145.87 (2C=C-N), 139.96, 137.91, 136.98, 134.88, 132.14, 130.87, 127.65, 126.65, 124.54, 120.65, 118.47, 99.19, 84.87, 50.65 (C-spiro), 46.38 (2CH_2_-piperidine), 45.48 (2CH_2_), 24.42, 22.94, 21.08 (3CH_2_-piperidine); MS (EI, 70 eV): *m*/*z* (%) = 693 (M^+^) (17.45%), 264 (100%); Anal. Calcd for C_35_H_31_N_7_O_7_S (693.74): C, 60.60; H, 4.50; N, 14.13; Found C, 60.86; H, 4.31; N, 14.26%.


**1′,9′-Bis(4-chlorobenzyl)-5-(piperidin-1-ylsulfonyl)-1′*H*-spiro[indoline-3,5′-pyrido[2,3-*d*:6,-5-*d*′]dipyrimidine]-2,2′,4′,6′,8′(3′*H*,7′*H*,9′*H*,10′*H*)-pentaone (4d)**


Yield 79% as light rose powder; M.P.: 327–329 °C; IR: ν/cm^−1^: 3364, 3171, 3109 (NH), 3050 (CH-Ar), 2940, 2850 (CH-Aliph.), 1773, 1670 (C=O); ^1^H NMR (400 MHz, DMSO) *δ*/ppm 11.90, 11.38, 11.15, 9.84 (4s, 4H, 4NH exchangeable by D_2_O), 7.59–7.55 (m, 3H, Ar-H), 7.44 (d, *J* = 8.8Hz, 1H, Ar-H), 7.40 (d, *J* = 8.8 Hz, 1H, Ar-H), 7.36 (d, *J* = 8.4 Hz, 2H, Ar-H), 7.32 (d, *J* = 7.6 Hz, 1H, Ar-H), 7.20 (d, 1H, Ar-H), 7.06 (d, *J* = 7.6 Hz, 2H, Ar-H), 5.28 (d, 2H, CH_2_), 5.18 (d, 2H, CH_2_), 2.81 (s, 4H, 2CH_2_-piperidine), 1.45 (s, 4H, 2CH_2_-piperidine), 1.32 (s, 2H, CH_2_-piperidine); ^13^C NMR (101 MHz, DMSO) *δ* 180.94, 172.03, 160.75, 157.84, 153.29 (5C=O), 150.15, 149.58 (2C=C-N), 146.75, 139.51, 133.94, 133.43, 131.70, 131.59, 130.49, 129.67, 129.57, 129.14, 128.92, 127.73, 127.63, 127.49, 125.49, 121.59, 117.54, 98.45, 83.90, 50.30 (C-spiro), 46.34 (2CH_2_-piperidine), 44.61 (2CH_2_), 24.46, 22.84, 21.06 (3CH_2_-piperidine); MS (EI, 70 eV): *m*/*z* (%) = 761 (M^+^) (37.72%), 763 (M^+2^) (24.23%), 765 (M^+4^) (32.28%), 336.97 (100%); Anal. Calcd for C_35_H_29_Cl_2_N_7_O_7_S (761.62): C, 55.12; H, 3.83; N, 12.86; Found C, 55.24; H, 3.93; N, 12.54%.


**1′,9′-Dimethyl-5-(piperidin-1-ylsulfonyl)-2′,8′-dithioxo-2′,3′,8′,9′-tetrahydro-1′*H*-spiro-[indoline-3,5′-pyrido[2,3-*d*:6,5-*d′*]dipyrimidine]-2,4′,6′(7′*H*,10′*H*)-trione (4e)**


Yield 76% as Buff powder; M.P.: 326–327 °C; IR: ν/cm^−1^: 3172, 3109 (NH), 3060 (CH-Ar), 2972, 2840 (CH-Aliph.), 1771, 1670 (C=O); ^1^H NMR (400 MHz, DMSO) *δ*/ppm 12.56, 12.45, 11.95, 9.94 (4s, 4H, 4NH exchangeable by D_2_O), 7.63 (dd, *J* = 8.4, 1.8 Hz, 1H, Ar-H), 7.58 (d, *J* = 8.4 Hz, 1H, Ar-H), 7.17 (d, 1H, Ar-H), 3.96, 3.74 (2s, 6H, 2CH_3_), 2.80 (s, 4H, 2CH_2_-piperidine), 1.43 (s, 4H, 2CH_2_-piperidine), 1.33 (s, 2H, CH_2_-piperidine); ^13^C NMR (101 MHz, DMSO) *δ* 176.20, 175.45 (2C=S), 161.56, 157.64, 155.10 (3C=O), 146.84, 139.37 (2C=C-N), 132.15, 130.95, 127.96, 125.28, 120.44, 118.11, 102.25, 88.26, 50.49 (C-spiro), 46.39 (2CH_2_-piperidine), 36.66, 36.34 (2CH_3_), 24.43, 22.93 (3CH_2_-piperidine); MS (EI, 70 eV): *m*/*z* (%) = 573 (M^+^) (34.33%), 135 (100%); Anal. Calcd for C_23_H_23_N_7_O_5_S_3_ (573.66): C, 48.16; H, 4.04; N, 17.09; Found C, 48.34; H, 3.88; N, 17.23%.


**5-(Morpholinosulfonyl)-1′*H*-spiro[indoline-3,5′-pyrido[2,3-*d*:6,5-*d′*]dipyrimidine]-2,2′,4′,-6′,8′(3′*H*,7′*H*,9′*H*,10′*H*)-pentaone (5a)**


Yield 81% as Off white powder; M.P.: 378–380 °C; IR: ν/cm^−1^: 3450, 3181 (NH), 3030 (CH-Ar), 2870 (CH-Aliph.), 1697, 1637 (C=O); ^1^H NMR (400 MHz, DMSO) δ/ppm 11.99, 11.26 (2s, 2H, 2NH exchangeable by D_2_O), 10.79 (s, 2H, 2NH exchangeable by D_2_O), 10.59, 9.68 (2s, 2H, 2NH exchangeable by D_2_O), 7.52 (dd, *J* = 8.4, 2.0 Hz, 1H, Ar-H), 7.28 (d, *J* = 8.4 Hz, 1H, Ar-H), 6.99 (d, 1H, Ar-H), 3.58 (t, 4H, 2CH_2_-O), 2.72 (s, 4H, 2CH_2_-N); ^13^C NMR (101 MHz, DMSO) δ/ppm 180.76, 161.90, 158.58, 152.30, 150.93 (5C=O), 149.87, 145.70 (2C=C-N), 139.75, 128.08, 127.84, 125.66, 121.85, 117.15, 97.26, 82.82, 65.17 (2CH_2_-O), 49.01 (C-spiro), 45.64 (2CH_2_-N); MS (EI, 70 eV): *m*/*z* (%) = 515 (M^+^) (20.15%), 129 (100%); Anal. Calcd for C_20_H_17_N_7_O_8_S (515.46): C, 46.60; H, 3.32; N, 19.02; Found C, 46.38; H, 3.51; N, 19.17%.


**1′,9′-Dimethyl-5-(morpholinosulfonyl)-1′*H*-spiro[indoline-3,5′-pyrido[2,3-*d*:6,5-*d′*]dipyrimi-dine]-2,2′,4′,6′,8′(3′*H*,7′*H*,9′*H*,10′*H*)-pentaone (5b)**


Yield 72% as light yellow powder; M.P.: 358–360 °C; IR: ν/cm^−1^: 3390, 3168 (NH), 3088 (CH-Ar), 2867 (CH.Aliph.), 1723, 1667 (C=O); ^1^H NMR (400 MHz, DMSO) *δ*/ppm 11.72, 11.08, 10.87, 9.77 (4s, 4H, 4NH exchangeable by D_2_O), 7.60 (d, *J* = 8.4 Hz, 1H, Ar-H), 7.55 (d, *J* = 8.4 Hz, 1H, Ar-H), 7.10 (s, 1H, Ar-H), 3.56 (s, 4H, 2CH_2_-O), 3.46 (s, 3H, CH_3_), 3.34–3.33 (m, 3H, CH_3_), 2.74 (s, 4H, 2CH_2_-N); ^13^C NMR (101 MHz, DMSO) *δ* 181.07, 160.51, 157.69, 153.56, 150.71 (5C=O), 149.92, 146.80 (2C=C-N), 140.06, 128.79, 127.84, 125.65, 121.74, 117.72, 97.94, 83.87, 65.20 (2CH_2_-O), 50.25 (C-spiro), 45.71 (2CH_2_-N), 30.37, 29.31 (2CH_3_); MS (EI, 70 eV): *m*/*z* (%) = 543 (M^+^) (21.48%), 228 (100%); Anal. Calcd for C_22_H_21_N_7_O_8_S (543.51): C, 48.62; H, 3.89; N, 18.04; Found C, 48.45; H, 3.99; N, 18.19%.


**1′,9′-Dibenzyl-5-(morpholinosulfonyl)-1′*H*-spiro[indoline-3,5′-pyrido[2,3-*d*:6,5-*d′*]dipyrimi-dine]-2,2′,4′,6′,8′(3′*H*,7′*H*,9′*H*,10′*H*)-pentaone (5c)**


Yield 77% as buff powder; M.P.: 244–246 °C; IR: ν/cm^−1^: 3323, 3170 (NH), 3058 (CH-Ar), 2984, 2923, 2840 (CH-Aliph.), 1725, 1653 (C=O); ^1^H NMR (400 MHz, DMSO) *δ*/ppm 11.92, 11.29, 11.04, 9.86 (s, 4H, 4NH exchangeable by D_2_O), 7.55 (dd, *J* = 8.4, 2.0 Hz, 1H, Ar-H), 7.38 (d, *J* = 7.6 Hz, 2H, Ar-H), 7.35 (d, *J* = 8.4 Hz, 2H, Ar-H), 7.31–7.30 (m, 2H, Ar-H), 7.28 (d, *J* = 6.8 Hz, 2H, Ar-H), 7.23 (t, *J* = 5.8 Hz, 3H, Ar-H), 7.05 (d, 1H, Ar-H), 5.00 (d, 2H, CH_2_), 4.89 (d, 2H, CH_2_), 3.52 (s, 4H, 2CH_2_-O), 2.65 (s, 4H, 2CH_2_-N); ^13^C NMR (101 MHz, DMSO) *δ* 180.97, 160.95, 157.66, 153.40, 150.28 (5C=O), 144.80 (2C=C-N), 139.64, 137.50, 136.01, 128.76, 128.30, 128.06, 127.62, 127.45, 127.11, 126.35, 125.66, 121.43, 117.38, 98.16, 82.63, 65.08 (2CH_2_-O), 50.24 (C-spiro), 45.62, 45.93 (2CH_2_) 43.62 (2CH_2_-N); MS (EI, 70 eV): *m*/*z* (%) = 695 (M^+^) (32.09%), 287 (100%); Anal. Calcd for C_34_H_29_N_7_O_8_S (695.71): C, 58.70; H, 4.20; N, 14.09; Found C, 58.49; H, 4.32; N, 14.21%.


**1′,9′-Bis(4-chlorobenzyl)-5-(morpholinosulfonyl)-1′*H*-spiro[indoline-3,5′-pyrido[2,3-*d*:6,5-*d′*]dipyrimidine]-2,2′,4′,6′,8′(3′*H*,7′*H*,9′*H*,10′*H*)-pentaone (5d)**


Yield 76% as light rose powder; M.P.: 347–349 °C; IR: ν/cm^−1^: 3257, 3188 (NH), 3092 (CH-Ar), 2926, 2858 (CH-Aliph.), 1720, 1640 (C=O); ^1^H NMR (400 MHz, DMSO) *δ*/ppm 11.91, 11.40, 11.15, 9.87 (4s, 4H, 4NH exchangeable by D_2_O), 7.59–7.57 (m, 1H, Ar-H), 7.55–7.52 (m, 1H, Ar-H), 7.48 (d, *J* = 8.8 Hz, 1H, Ar-H), 7.41 (d, *J* = 7.2 Hz, 1H, Ar-H), 7.37 (d, *J* = 6.2 Hz, 1H, Ar-H), 7.34 (d, *J* = 8.0 Hz, 1H, Ar-H), 7.29 (d, *J* = 7.6 Hz, 1H, Ar-H), 7.25 (d, 1H, Ar-H), 7.19 (d, 1H, Ar-H), 7.06 (d, *J* = 7.6 Hz, 1H, Ar-H), 6.89 (d, 1H, Ar-H), 5.22 (d, 2H, CH_2_), 5.05 (d, 2H, CH_2_), 3.57 (s, 4H, 2CH_2_-O), 2.76 (s, 4H, 2CH_2_-N); ^13^C NMR (101 MHz, DMSO) *δ* 180.98, 160.76, 157.87, 157.72, 150.15 (5C=O), 149.56, 146.77 (2C=C-N), 139.91, 134.31, 133.94, 133.45, 133.31, 131.68, 131.57, 129.57, 129.33, 129.05, 128.48, 128.03, 127.53, 127.21, 126.21, 125.67, 121.73, 117.65, 98.37, 83.98, 65.18 (2CH_2_-O), 50.31 (C-spiro), 45.67 (2CH_2_-O), 44.66, 43.08 (2CH_2_); MS (EI, 70 eV): *m*/*z* (%) = 763 (M^+^) (46.23%), 765 (M^+2^) (18.75%), 767 (M^+4^) (23.17%), 738.78 (100%); Anal. Calcd for C_34_H_27_Cl_2_N_7_O_8_S (763.59): C, 53.41; H, 3.56; N, 12.82; Found C, 53.21; H, 3.75; N, 12.98%.


**1′,9′-Dimethyl-5-(morpholinosulfonyl)-2′,8′-dithioxo-2′,3′,8′,9′-tetrahydro-1′*H*-spiro[indo-line-3,5′-pyrido[2,3-*d*:6,5-*d′*]dipyrimidine]-2,4′,6′(7′*H*,10′*H*)-trione (5e)**


Yield 74% as light yellow powder; M.P.: 341–343 °C; IR: ν/cm^−1^: 3240, 3186 (NH), 3055 (CH-Ar), 2932, 2860 (CH-Aliph.), 1756, 1673 (C=O); ^1^H NMR (400 MHz, DMSO) *δ*/ppm 12.57, 12.44, 11.96, 9.98 (4s, 4H, 4NH exchangeable by D_2_O), 7.64 (dd, *J* = 8.4, 1.8 Hz, 1H, Ar-H), 7.61 (d, *J* = 8.4 Hz, 1H, Ar-H), 7.19 (s, 1H, Ar-H), 3.97, 3.74 (2s, 6H, 2CH_3_), 3.56 (s, 4H, 2CH_2_-O), 2.76 (s, 4H, 2CH_2_-N); ^13^C NMR (101 MHz, DMSO) *δ* 180.07, 176.20 (2C=S), 175.44, 157.61, 155.11 (3C=O), 153.60, 146.82 (2C=C-N), 139.77, 129.42, 128.23, 125.65, 120.55, 118.24, 102.04, 88.32, 65.17 (2CH_2_-O), 50.48 (C-spiro), 45.71 (2CH_2_-N), 36.69, 36.35 (2CH_3_); MS (EI, 70 eV): *m*/*z* (%) = 575 (M^+^) (39.80%), 566 (100%); Anal. Calcd for C_22_H_21_N_7_O_6_S_3_ (575.63): C, 45.90; H, 3.68; N, 17.03; Found C, 45.71; H, 3.54; N, 17.17%.

### 3.2. Antiviral Activity

#### 3.2.1. Cytotoxicity Assay

Samples were diluted with Dulbecco’s Modified Eagle Medium (DMEM). Stock solutions of the test compounds were prepared in 10% DMSO in dd H_2_O. The cytotoxic activity of the extracts were tested in ATCC Vero E6 cells by using the 3-(4, 5-dimethylthiazol -2-yl)-2, 5-diphenyltetrazolium bromide (MTT) method [26] with minor modification. Briefly, the cells were seeded in 96-well plates (100 µL/well at a density of 3 × 10^5^ cells/mL) and incubated for 24 h at 37 °C in 5% CO_2_. After 24 h, cells were treated with various concentrations of the tested compounds in triplicates. After a further 24 h, the supernatant was discarded, and cell monolayers were washed with sterile phosphate buffer saline (PBS) three times and MTT solution (20 µL of 5 mg/mL stock solution) was added to each well and incubated at 37 °C for 4 h followed by medium aspiration. In each well, the formed formazan crystals were dissolved with 200 µL of acidified isopropanol (0.04 M HCl in absolute isopropanol = 0.073 mL HCl in 50 mL isopropanol). Absorbance of formazan solutions were measured at λ_max_ 540 nm with 620 nm as a reference wavelength using a multi-well plate reader. The percentage of cytotoxicity compared to the untreated cells was determined with the following equation.
$$\%\;\mathrm{Cytotoxicity}=\frac{\left(\mathrm{absorbance}\;\mathrm{of}\;\mathrm{cells}\;\mathrm{without}\;\mathrm{treatment}-\mathrm{absorbace}\;\mathrm{of}\;\mathrm{cells}\;\mathrm{with}\;\mathrm{treatment}\;\right)\times 100}{\mathrm{absorbance}\;\mathrm{of}\;\mathrm{cells}\;\mathrm{without}\;\mathrm{treatment}}$$

The plot of % cytotoxicity versus sample concentration was used to calculate the concentration, which exhibited 50% cytotoxicity (IC_50_). Three independent experiments for each concentration were performed.

#### 3.2.2. Plaque Reduction Assay

Assay was carried out according to the method of [37] in a six-well plate where Vero E6 cells (10^5^ cells/mL) were cultivated for 24 h at 37 °C. Cells were grouped into negative and positive control groups without treatment and treated groups. Severe Acute Respiratory Syndrome Coronavirus (SARS-CoV-2) virus (isolated from Egyptian patients with the help of Egy-Army with GenBank code (MT776904) and the link of this on-site represented as (https://www.ncbi.nlm.nih.gov/nuccore/MT776904, accessed on 5 February 2022) was diluted to give 10^3^ PFU/well and mixed with the safe concentration of the tested compounds and incubated for 1 h at 37 °C before being added to the cells. Growth medium was removed from the cell culture plates and the cells were inoculated (100 µL/well) with the virus with the tested compounds. After 1 h contact time for virus adsorption, 3 mL of DMEM supplemented with 2% agarose was added and the tested compounds were then added onto the cell monolayer. Plates were left to solidify and were incubated at 37 °C until formation of viral plaques (3 to 4 days) appeared. Formalin (10%) was added for two hours before plates were stained with 0.1% crystal violet in distilled water. Control wells were included where the untreated virus was incubated with Vero E6 cells. Finally, the plaques were counted and the percentage reduction in plaques formation (% Reduction) in comparison to control wells was recorded according to the following Equation (1):(1)$$\mathrm{Reduction}\;\left(\%\right)=\frac{\mathrm{Viral}\;\mathrm{count}\;\left(\mathrm{untreated}\right)-\mathrm{Viral}\;\mathrm{count}\left(\mathrm{treated}\right)}{\mathrm{Viral}\;\mathrm{count}\left(\mathrm{untreated}\right)\times 100\;}$$
where the viral count (untreated) is the viral count in wells where the virus was untreated with the compounds. Additionally, the viral count (treated) is the viral count in wells where the virus was treated with the compounds. Additionally, this performance is shown in Table 2, which presents the inhibition percentage with three independent experiments for each concentration.

Enzyme assay was prepared according to reported method [38].

### 3.3. Molecular Docking Study

Firstly, the most active derivatives **3a**, **4b**, **4d**, **4e** and positive controls (**Chloroquine** and **Remdesivir**) were built using ChemBioDraw 2014, before being exported to Molecular Operating Environmental 10.2008 (MOE) [39,40]. Additionally, the structure was prepared by protonation, and the energy was minimized using the MMFF94x forcefield. The crystal structure of different proteins as a structure of RNA-dependent RNA polymerase (RdRp) (PDB: 6M71) and spike glycoprotein (SGp) (PDB: 6VXX) were obtained from the protein data bank [41,42]. The selected protein and docking process structure was prepared according to the default protocol and according to the previously reported method [43] using the dummy atoms to generate the active site. We selected only one chain for SARS-CoV-2 spike glycoprotein (PDB: 6VXX) (Chain A). Additionally, the SARS-CoV-2 RNA-dependent RNA polymerase (PDB: 6M71) chain A was selected for the docking proposed. The docking simulation inside the active site was performed using the trigonal matcher placement method using London DG as a scoring function. Its energy was represented by Kcal/mol and the top-scoring pose was inspected visually.

### 3.4. Computational Study

The geometrical optimization of the most active derivatives **3a**, **4b**, **4d**, **4e** and positive controls (**Chloroquine** and **Remdesivir)** were performed using DFT methods through the Gaussian 09 set of the program according to previously reported methods [32]. The global descriptors of chemical reactivity are related to frontier molecular orbital and are calculated simply depending on previously reported methods [44,45]. The global descriptors used in this study are: *IP* =ionization potential, *EA* = Electron affinity, *X* = Electronegativity, *ɳ* = Chemical hardness, *S* = Chemical softness, *µ* = Chemical potential, *ω* = Electrophilic index.

## 4. Conclusions

In summary, a series of newly synthesized compounds of spirooxindole based on uracil derivatives **3a**–**d, 4a**–**e**, and **5a**–**e** were prepared through a simple method by heating 5-(substituted-1-ylsulfonyl)indoline-2,3-dione **1a**–**c** under reflux condition with 6-aminouracil derivatives **2a**–**e**. Using the MTT assay, the new hybrids were investigated for anti-viral activity against SARS-CoV-2. As a result, the majority of the compounds tested had moderate to strong anti-SARS-CoV-2 action. In addition, all new compounds were tested for percentage of inhibition using the plaque reduction assay, which revealed that compounds **3a**, **4b**, **4d**, and **4e** had a high range of action. The mechanism of action on RNA-dependent RNA polymerase (RdRp) and spike glycoprotein was studied further with these four hybrids. The results were quite promising, as the new hits were found to be as equally effective as chloroquine, which was previously used to treat COVID-19. These findings motivate our team to conduct more advanced studies on the novel hybrids in order to learn more about the mechanism of action in future investigations.

## Data Availability

Data is contained within the article and Appendix A.

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
