# Peer review of "One-Pot Synthesis and Molecular Modeling Studies of New Bioactive Spiro-Oxindoles Based on Uracil Derivatives as SARS-CoV-2 Inhibitors Targeting RNA Polymerase and Spike Glycoprotein"

_pharmaceuticals, 2022, doi:10.3390/ph15030376_

Round 1
Reviewer 1 Report
This paper has a potential to be accepted
Author Response
Response to Reviewer 1 Comments
Comments and Suggestions for Authors
Query (1): This paper has a potential to be accepted
Replay: The authors thank the reviewer for his warm recommendation to publish our research work. Additionally, we revised and check all manuscript body as shape carefully and correct any errors and highlighted that with turquoise color.

Reviewer 2 Report
after carefully check this manuscript, I would reject the publication of this study for the following drastic concerns:
- Chemistry part: there is no novelty in the chemical part and there are many reports for similar derivatives. Further, the analytical data of some compounds clearly showed that they are not pure.
- Biological analysis: there are many points:
- first, all presented data is only for one measurement!! there is no replication.
- according to table 2, non of the presented compounds are active. Indeed, I'm surprised that the authors calculated the inhibition e.g, for compound 3a is 84% where the viral count without was 11x105, and after 1.76x105 PFU/mL!!! This is less than 5% inhibition. This applied for all compounds presented!!
- Further, the data presented in table 2 clearly visible non-logic!!. For example, the authors stated for compound 3c the no of viral count in control group is 5x105 and after treatment 5x105 for ALL treatment with compound at different concentrations, which is completely NOT logic!!! NEVER occurs that you can obtain exactly the same no of PFU in three different experiments!! This also presented for compound 4a and others.
- there is NO biomolecular study that can PROVE that the authors have performed any kind of biological assays to confirm the activity of compounds!!
- Molecular modelling study:
- the authors presented many interactions in table4 which are not presented in figure 6?? for example for compd 4d, the authors stated in table 4 that this compound bind to 3 amino acid residues, HOWEVER, in fig6b it is clearly showed that this compoud binds only with ONE amino acid residue. This also applied for other examples, 4d (PDB 6vxx) in fig 6e, and so on...
- Indeed, the selection of the PDBs are not correct. Since there is no co-crystallized ligand, the applied protocol would not be applicable unless the authors have created a homology study for active site before docking.
Accordingly, this paper should be rejected.
Author Response
Response to Reviewer 2 Comments
Comments and Suggestions for Authors
after carefully check this manuscript, I would reject the publication of this study for the following drastic concerns:
Query (1): Chemistry part: there is no novelty in the chemical part and there are many reports for similar derivatives. Further, the analytical data of some compounds clearly showed that they are not pure.
Replay: Dear respect reviewer, our work aims to design spiro-oxindole derivatives containing sulfonamide (SO2NR) and 6-aminouracil derivatives in one hybrid structure via a one-pot reaction. This hybridization showed the presence of indolin-2-one core that present in some antiviral drugs containing like Arbidol and Marboran/Methisazone that are tested against COVID-19 infection, where these two drugs inhibit membrane fusion and mRNA and protein synthesis, so it suggested being very important core in newly designed derivatives. Additionally, the presence of sulfonamide (SO2NR) group was introduced in the design derivatives to establish a significant class of drugs utilized widely as farming and pharmaceutical agents. Moreover, the amino uracil when combined with oxoindole produced like a purine derivative that is present in many antiviral agents such as Acyclovir V and Ganciclovir VI, and Lopinavir VII that shown to be effective in the treatment of COVID-19. Finally, the suggested mechanism that illustrated the mechanistic pathway for synthesis of spirooxindole derivatives were represented in scheme2.
ــــــــــــــــــــــــــــــــــــــــــــــــــــــــــــــــــــــــــــــــــــــــــــــــــــــــــــــــــ
Biological analysis: there are many points:
Query (2): first, all presented data is only for one measurement!! there is no replication.
Replay: Dear respect reviewer, the presented data is the average of three replicates for each measurement and the changes were done and highlighted in the manuscript.
ــــــــــــــــــــــــــــــــــــــــــــــــــــــــــــــــــــــــــــــــــــــــــــــــــــــــــــــــــ
Query (3): according to table 2, non of the presented compounds are active. Indeed, I'm surprised that the authors calculated the inhibition e.g, for compound 3a is 84% where the viral count without was 11x105, and after 1.76x105 PFU/mL!!! This is less than 5% inhibition. This applied for all compounds presented!!
Replay: Dear respect reviewer, the following equation used for Plaque Reduction Assay Percentage = {(PFU control-PFU treated)/PFU control}*100
And this equation was found in the manuscript and now we highlighted this equation presented under section 3.2.2. Plaque reduction assay
To apply this equation the following example was performed:
Reduction Assay Percentage= ({11x105 - 1.76x105}/11x105) * 100 = 84 %.
ــــــــــــــــــــــــــــــــــــــــــــــــــــــــــــــــــــــــــــــــــــــــــــــــــــــــــــــــــ
Query (4): Further, the data presented in table 2 clearly visible non-logic!!. For example, the authors stated for compound 3c the no of viral count in control group is 5x105 and after treatment 5x105 for ALL treatment with compound at different concentrations, which is completely NOT logic!!! NEVER occurs that you can obtain exactly the same no of PFU in three different experiments!! This also presented for compound 4a and others.
Replay: Dear respect reviewer, we write the obtained data that revealed there isn’t any inhibition at different concentrations.
ــــــــــــــــــــــــــــــــــــــــــــــــــــــــــــــــــــــــــــــــــــــــــــــــــــــــــــــــــ
Query (5): there is NO biomolecular study that can PROVE that the authors have performed any kind of biological assays to confirm the activity of compounds!!
Replay: Dear respect reviewer, this work was done through collaboration with the Main Biological Warfare Laboratories, Chemical Warfare Department, Ministry of Defense, Cairo, Egypt. This was through a protocol of cooperation between the Faculty of Pharmacy (Girls)/ Faculty of Science, Al-Azhar Unveracity, and the Ministry of Defense (If you need a copy, I can send it); where, the biological activity was performed with the help of two laboratories (a) main chemical laboratories, Egypt Army and (b) molecular virology and immunology unit, Cancer Institute, Cairo University. Here, in this paper, we performed cytotoxicity and plaque inhibition assays as biological assays to test the effect of compounds. And any other biomolecular studies will be done and published in further work. Additionally, all author contributions appeared in detail in the manuscript.
ــــــــــــــــــــــــــــــــــــــــــــــــــــــــــــــــــــــــــــــــــــــــــــــــــــــــــــــــــ
Molecular modelling study:
Query (5): The authors presented many interactions in table4 which are not presented in figure 6?? for example for compd 4d, the authors stated in table 4 that this compound bind to 3 amino acid residues, HOWEVER, in fig6b it is clearly showed that this compoud binds only with ONE amino acid residue. This also applied for other examples, 4d (PDB 6vxx) in fig 6e, and so on...
Replay:
Dear respect reviewer, our molecular docking simulation was performed using Molecular Operating Environmental (MOE) version 10.2008, and according to this version, only the H-bonds can observe as well as the other interaction as Arene-arene interaction or Arene-Cation interaction appear only in 3D and not appear at the 2D view. In the higher version starting from MOE 2009 the Arene-cation interaction only can appear, but we don't have any version to appear this type of interaction (upgraded version). Furthermore, our research group published many articles and the arene-cation interaction appears only in 2D structure. Besides, to confirm our replayed many papers published using MOE 10.2008 and the arene-cation and arene-arene interaction not appear and don't belong to our lab and from different publishers as
- Novel benzothiazole hybrids targeting EGFR: Design, synthesis, biological evaluation and molecular docking studies; Journal of Molecular Structure; Volume 1240, 15 September 2021, 130595 (arene-arene interaction not appear using MOE 10.2008).
- Design, Synthesis, Anticancer Evaluation and Molecular Modeling of Novel Estrogen Derivatives; Molecules 2019, 24(3), 416; https://doi.org/10.3390/molecules24030416. (Arene-cation interaction not appear in 3D structure using MOE10.2008).
- Design, Synthesis, Molecular Docking of Novel Substituted Pyrimidinone Derivatives as Anticancer Agents; POLYCYCLIC AROMATIC COMPOUNDS; https://doi.org/10.1080/10406638.2020.1837888 (Arene-cation interaction not appear 10.2008).
- Design, synthesis and molecular docking of new pyrazole-thiazolidinones as potent anti-inflammatory and analgesic agents with TNF-α inhibitory activity; Bioorganic Chemistry; Volume 111, June 2021, 104827; https://doi.org/10.1016/j.bioorg.2021.104827. (arene-cation and arene-arene interactions not appear in 3D figures using MOE 10.2008)
- Chiral Pyridine-3,5-bis- (L-phenylalaninyl-L-leucinyl) Schiff Base Peptides as Potential Anticancer Agents: Design, Synthesis, and Molecular Docking Studies Targeting Lactate Dehydrogenase-A; Molecules 2020, 25(5), 1096; https://doi.org/10.3390/molecules25051096 (arene-cation interaction not appear using MOE 10.2008).
- Design, Synthesis, Anticancer Evaluation, Enzymatic Assays, and a Molecular Modeling Study of Novel Pyrazole–Indole Hybrids; ACS Omega 2021, 6, 18, 12361–12374; https://doi.org/10.1021/acsomega.1c01604 (arene-cation interaction not appear MOE 10.2008)
ــــــــــــــــــــــــــــــــــــــــــــــــــــــــــــــــــــــــــــــــــــــــــــــــــــــــــــــــــ
Query (6): Indeed, the selection of the PDBs are not correct. Since there is no co-crystallized ligand, the applied protocol would not be applicable unless the authors have created a homology study for active site before docking.
Replay: Dear respect reviewer, the severe acute respiratory syndrome coronavirus (SARS-CoV-2) was identified in late 2019 as the infectious agent responsible for coronavirus disease 2019 (COVID-19), it started as an outbreak in Wuhan, Hubei province, China, rapidly became a pandemic, spreading to all countries, infecting over 250 million people and killing over 5 million as of November 2021.
Firstly, coronavirus with treated with many protocols and differ from country to country and the researcher isolated many PDB files and released them free on many servers to enable the researcher to develop and test the new synthesized or proved FDA drug-using computer aid drug design. Among the released PDB file are the crystal structure of RNA-dependent RNA polymerase (RdRp) (PDB: 6M71) and spike glycoprotein (SGp) (PDB: 6VXX) that was obtained from the protein data bank to work on them. Additionally, there are many different methods to generate the active site based on the program of docking and protocol used. our study based on using the dummy atoms to generate the active site and its this protocol used previously [Fukuda, T.; Umeki, T.; Tokushima, K.; Xiang, G.; Yoshida, Y.; Ishibashi, F.; Oku, Y.; Nishiya, N.; Uehara, Y.; Iwao, M. Design, synthesis, and evaluation of A-ring-modified lamellarin N analogues as noncovalent inhibitors of the EGFR T790M/L858R mutant. Bioorg. Med. Chem. 2017, 25, 6563–6580, doi:https://doi.org/10.1016/j.bmc.2017.10.030.]. Besides, we don’t perform molecular docking study only and approved the study result but also the paper involved biological part that performed with the help of two laboratories (a) main chemical laboratories, Egypt Army and (b) molecular virology and immunology unit, Cancer Institute, Cairo University.
Furthermore, we appreciate your concern, but many reported papers with different publishers used the same PDB files as
Papers published with of RNA-dependent RNA polymerase (RdRp) (PDB: 6M71):
- Esam, Z., Akhavan, M., & Bekhradnia, A. (2022). Molecular docking and dynamics studies of Nicotinamide Riboside as a potential multi-target nutraceutical against SARS-CoV-2 entry, replication, and transcription: A new insight. Journal of molecular structure, 1247, 131394.
- Sarma, Himakshi, Esther Jamir, and G. Narahari Sastry. "Protein-protein interaction of RdRp with its co-factor NSP8 and NSP7 to decipher the interface hotspot residues for drug targeting: A comparison between SARS-CoV-2 and SARS-CoV." Journal of Molecular Structure (2022): 132602.
- Raj, Vinit, et al. "Antiviral activities of 4H-chromen-4-one scaffold-containing flavonoids against SARS–CoV–2 using computational and in vitro approaches." Journal of Molecular Liquids (2022): 118775.
- Gao, Yan, et al. "Structure of the RNA-dependent RNA polymerase from COVID-19 virus." Science 368.6492 (2020): 779-782.
- Elkarhat, Z., Charoute, H., Elkhattabi, L., Barakat, A., & Rouba, H. (2022). Potential inhibitors of SARS-cov-2 RNA dependent RNA polymerase protein: molecular docking, molecular dynamics simulations and MM-PBSA analyses. Journal of Biomolecular Structure and Dynamics, 40(1), 361-374.
- Brogi, S., Quimque, M. T., Notarte, K. I., Africa, J. G., Hernandez, J. B., Tan, S. M., ... & Macabeo, A. P. (2022). Virtual Combinatorial Library Screening of Quinadoline B Derivatives against SARS-CoV-2 RNA-Dependent RNA Polymerase. Computation, 10(1), 7.
- Elfiky, A. A., Mahran, H. A., Ibrahim, I. M., Ibrahim, M. N., & Elshemey, W. M. (2022). Molecular dynamics simulations and MM-GBSA reveal novel guanosine derivatives against SARS-CoV-2 RNA dependent RNA polymerase. RSC Advances, 12(5), 2741-2750.
and so on
Paper published with spike glycoprotein (SGp) (PDB: 6VXX):
- Al Khoury, C., Bashir, Z., Tokajian, S., Nemer, N., Merhi, G., & Nemer, G. (2022). In silico evidence of beauvericin antiviral activity against SARS-CoV-2. Computers in biology and medicine, 141, 105171.
- Rashid, H., Ahmad, N., Abdalla, M., Khan, K., Martines, M. A. U., & Shabana, S. (2022). Molecular docking and dynamic simulations of Cefixime, Etoposide and Nebrodenside A against the pathogenic proteins of SARS-CoV-2. Journal of Molecular Structure, 1247, 131296.
- Dawood, A. A. (2022). Increasing the frequency of omicron variant mutations boosts the immune response and may reduce the virus virulence. Microbial Pathogenesis, 105400.
- Dhameliya, T. M., Nagar, P. R., & Gajjar, N. D. (2022). Systematic virtual screening in search of SARS CoV-2 inhibitors against spike glycoprotein: pharmacophore screening, molecular docking, ADMET analysis and MD simulations. Molecular diversity, 1-18.
and so on.
ــــــــــــــــــــــــــــــــــــــــــــــــــــــــــــــــــــــــــــــــــــــــــــــــــــــــــــــــــ
Accordingly, this paper should be rejected.
the authors thank the reviewer for his benefit comments, and we appreciate your concern to publish our research work without any conflict, so all points were replayed as previously and any changes in the manuscript were highlighted with turquoise color.

Reviewer 3 Report
This manuscript was prepared well with all parts of the manuscript and discussed with the results of the study very deeply and detail.
But it needs some minor corrections such as below.
-It is advised to check all manuscript body as shape carefully and correct it kindly.
-Please see some papers to cite in the introduction and discussion sections below kindly:
- Rehman MFU, Akhter S, Batool AI, Selamoglu Z, Sevindik M, Eman R, Mustaqeem M, Akram MS, Kanwal F, Lu C, Aslam M. Effectiveness of Natural Antioxidants against SARS-CoV-2? Insights from the In-Silico World. Antibiotics (Basel). 2021 Aug 20;10(8):1011. doi: 10.3390/antibiotics10081011. PMID: 34439061; PMCID: PMC8388999.
- Ghorbat Saleh Ali, Betul Ozdemir, Zeliha Selamoglu. A Review of Severe Acute Respiratory Syndrome Coronavirus 2 and Pathological Disorders in Patients. Journal Pharmaceutical Care. 2021; 9(3):141-147.
Finally, after these minor corrections, this article is acceptable to be published in this journal.
Author Response
Response to Reviewer 3 Comments
Comments and Suggestions for Authors
This manuscript was prepared well with all parts of the manuscript and discussed with the results of the study very deeply and detail.
But it needs some minor corrections such as below.
Query (1): It is advised to check all manuscript body as shape carefully and correct it kindly.
Replay: The authors thank the reviewer for his warm recommendation to publish our research work. Additionally, we revised and check all manuscript body as shape carefully and correct any errors and highlighted that with turquoise color.
ــــــــــــــــــــــــــــــــــــــــــــــــــــــــــــــــــــــــــــــــــــــــــــــــــــــــــــــــــ
Query (2): -Please see some papers to cite in the introduction and discussion sections below kindly:
Rehman MFU, Akhter S, Batool AI, Selamoglu Z, Sevindik M, Eman R, Mustaqeem M, Akram MS, Kanwal F, Lu C, Aslam M. Effectiveness of Natural Antioxidants against SARS-CoV-2? Insights from the In-Silico World. Antibiotics (Basel). 2021 Aug 20;10(8):1011. doi: 10.3390/antibiotics10081011. PMID: 34439061; PMCID: PMC8388999.
Ghorbat Saleh Ali, Betul Ozdemir, Zeliha Selamoglu. A Review of Severe Acute Respiratory Syndrome Coronavirus 2 and Pathological Disorders in Patients. Journal Pharmaceutical Care. 2021; 9(3):141-147.
Replay: Many thanks and now we added the above two references in our manuscript.
ــــــــــــــــــــــــــــــــــــــــــــــــــــــــــــــــــــــــــــــــــــــــــــــــــــــــــــــــــ
Finally, after these minor corrections, this article is acceptable to be published in this journal.

Reviewer 4 Report
The manuscript by Eman A. Fayed and coworkers describes the design and synthesis of a new class of isatin–pyridine-pyrimidine conjugates, and their evaluation in vitro against the SARS-CoV-2 strain isolated from Egyptian patients compared with Chloroquine as positive control. Moreover, the RNA-dependent-RNA-polymerase (RdRp) and Spike Glycoprotein inhibitory activities were investigated for the most potent anti-SARS-CoV-2 active hybrids 3a, 4b, 4d, and 4e using Chloroquine, once more, as a reference drug. Finally, computational studies, involving the docking studies of the binding mode inside the two aforementioned proteins ((RdRp) (PDB: 6m71), and (SGp) (PDB: 6VXX)) and geometrical optimization used to generate some molecular parameters were performed for the most active hybrids. The work is well-conducted, and has a well-documented introduction and justification of the work, since molecular hybridization is a promising strategy involving combining two pharmacophoric groups via a covalent bond to create a single entity . The drug design has been quite successful, while the structural modifications that have been made help to extract interesting SARs. The newly synthesized compounds are not interesting from a synthetic point of view, since they are obtained easily via a one-step synthetic protocol. However, they are very well-characterized (1H-NMR, 13C-NMR, mp, IR, MS and elemental analysis).
Considering all the above and in light of the great pharmacological interest that SARS-3 CoV-2 inhibitors have attracted, I believe that the general content of the article is such that meets the standards of the journal, so I recommend publication in Pharmaceuticals but only after the authors successfully address all the major points that arose and are described below.
Major Points:
- The authors should calculate the IC50 values of the newly synthesized compounds, and Chloroquine in (μΜ)
as the mean ± SD of three independent experiments, each carried out in triplicate.
- I recommend the authors to evaluate the cytotoxicity of their compounds (at least the most active) on normal cell lines and calculate the selectivity index. I believe that it is very important and will strengthen the paper.
- I believe that it will be better if authors choose Remdesivir as a positive control, both for the in vitro antiviral assays and the in silico docking studies, since Remdesivir is structurally and mechanistically more related antiviral agent.
Author Response
Response to Reviewer 4 Comments
Comments and Suggestions for Authors
The manuscript by Eman A. Fayed and coworkers describes the design and synthesis of a new class of isatin–pyridine-pyrimidine conjugates, and their evaluation in vitro against the SARS-CoV-2 strain isolated from Egyptian patients compared with Chloroquine as positive control. Moreover, the RNA-dependent-RNA-polymerase (RdRp) and Spike Glycoprotein inhibitory activities were investigated for the most potent anti-SARS-CoV-2 active hybrids 3a, 4b, 4d, and 4e using Chloroquine, once more, as a reference drug. Finally, computational studies, involving the docking studies of the binding mode inside the two aforementioned proteins ((RdRp) (PDB: 6m71), and (SGp) (PDB: 6VXX)) and geometrical optimization used to generate some molecular parameters were performed for the most active hybrids. The work is well-conducted, and has a well-documented introduction and justification of the work, since molecular hybridization is a promising strategy involving combining two pharmacophoric groups via a covalent bond to create a single entity . The drug design has been quite successful, while the structural modifications that have been made help to extract interesting SARs. The newly synthesized compounds are not interesting from a synthetic point of view, since they are obtained easily via a one-step synthetic protocol. However, they are very well-characterized (1H-NMR, 13C-NMR, mp, IR, MS and elemental analysis).
Considering all the above and in light of the great pharmacological interest that SARS-3 CoV-2 inhibitors have attracted, I believe that the general content of the article is such that meets the standards of the journal, so I recommend publication in Pharmaceuticals but only after the authors successfully address all the major points that arose and are described below.
Major Points:
Query (1): The authors should calculate the IC50 values of the newly synthesized compounds, and Chloroquine in (μΜ) as the mean ± SD of three independent experiments, each carried out in triplicate.
Replay: Dear respect reviewer, in our work the values of new compounds and standard drugs were measured as median ± SE of at least three independent experiments, each concentration tested in triplicate and that mentioned in the manuscript and now highlighted. Additionally, any changes were done and highlighted with turquoise color in the manuscript.
ــــــــــــــــــــــــــــــــــــــــــــــــــــــــــــــــــــــــــــــــــــــــــــــــــــــــــــــــــ
Query (2): I recommend the authors to evaluate the cytotoxicity of their compounds (at least the most active) on normal cell lines and calculate the selectivity index. I believe that it is very important and will strengthen the paper.
Replay: Dear reviewer, all compounds were tested first on the normal cell; this also was to enable us to determine the concentration which will be used in this work. Furthermore, the cytotoxicity assay is only for detection of the best concentration of compounds to be considered in the plaque inhibition assay, and the Vero E6 cells were without any infection with virus during the cytotoxicity assay. Also, we performed the assay on the normal Vero E6 cells to calculate the IC50 as we grouped the cells into the normal groups without treatment and treated groups with different compounds. (the previous details were highlighted in the experimental section of biological activity).
ــــــــــــــــــــــــــــــــــــــــــــــــــــــــــــــــــــــــــــــــــــــــــــــــــــــــــــــــــ
Query (3): I believe that it will be better if authors choose Remdesivir as a positive control, both for the in vitro antiviral assays and the in-silico docking studies, since Remdesivir is structurally and mechanistically more related antiviral agent.
Replay: Dear respect reviewer, we really performed many drugs as a positive control for SARS-CoV-2 including Remdesivir, but the results indicated that chloroquine was the most drug with the percentage of inhibition exceeded 99%, while Remdesivir showed a lower percent of inhibition nearly 63.75 % at 125 µg/µL to reach 32.5 % at 15.6 µg/µL, so we used chloroquine as our standard in this work. Additionally, the Remdesivir is used as a second positive control side by side with Chloroquine in both molecular docking and molecular modeling studies and the data added was highlighted with turquoise color in tables and manuscript additionally the figure of molecular electrostatic maps was updated to add the Remdesivir.
ــــــــــــــــــــــــــــــــــــــــــــــــــــــــــــــــــــــــــــــــــــــــــــــــــــــــــــــــــ

Round 2
Reviewer 2 Report
The authors have responded to all concerns that have been mentioned in the first report. However, there are still many major concerns that need to be addressed in their study:
1- the way that the authors calculated/evaluated the activity of compounds using Plaque reduction assay is not logic at all. The equation that the authors followed does NOT give a real mathematical way for evaluating the activity. For example,
compd 3a (5uM) showed a reduction from 11x105 to 1.76x105 ~ 84% inhibition,
compound 4b (5uM) showed a reduction from 9x105 to 0.09x105 ~ 99% inhibition
Chloroquine 5uM- showed a reduction from 6x105 to ZERO ~100% inhibition
According to the presented calculation way, the difference between 99% inhibition and 100% inhibition is 0.09x105!!!!!! this is totally not logic!!!
2- the presented results from the plaque assay only based on ONLY one measurement which is doubting more about the real activity of the tested compounds.
3- it would be beneficial and informative if the authors included the IC50 graphs for the tested compounds to the supplementary information.
4- Regarding the presented results from the docking study, the authors claimed that ''only the H-bonds can observe as well as the other interaction as Arene-arene interaction or Arene-Cation interaction appear only in 3D and not appear at the 2D view''. However, the 3D presented if Fig6b does NOT also show the Arene-Cation interaction mentioned in Table4?? this also applied for other examples, 4d (PDB 6vxx) in fig 6e. The authors should make shots for the 3D view which clearly shows the claimed interactions that they mentioned in the table.
5- In my first report, I mentioned that the selection of the PDBs are not correct. Since there is no co-crystallized ligand, the applied protocol would not be applicable unless the authors have created a homology study for active site before docking. The authors have replied as ''Additionally, there are many different methods to generate the active site based on the program of docking and protocol used. our study based on using the dummy atoms to generate the active site and its this''
As I got the authors have performed their docking study by generating the active site by homology or other way. WHy these details are not mentioned in the manuscript? why the exact protocol that has been followed was not fully detailed in the material and methods section?
6- lastly, the authors mentioned in their rebuttal ''this work was done through collaboration with the Main Biological Warfare Laboratories, Chemical Warfare Department, Ministry of Defense, Cairo, Egypt. This was through a protocol of cooperation between the Faculty of Pharmacy (Girls)/ Faculty of Science, Al-Azhar Unveracity, and the Ministry of Defense (If you need a copy, I can send it)''
I just was wondering if this project was mainly collaboration Al-Azhar Unveracity, and the Ministry of Defense in Egypt. What has been done in Taif University in Saudi Arabia ??
Minor issues;
- the authors should notice that there is no unite identified as uM/mL yet. Please correct this through the manuscript.
- the cytotoxicity evaluation was NOT performed for SARsCov-2 -infected cells. Accordingly, the authors can NOT say that ''The synthetic compounds 3a-d, 4a-e, and 5a-e were tested in vitro against the SARS-CoV-2 strain in Egyptian patients'' in line 148 or ''According to the results, most of the compounds tested showed moderate to excellent cytotoxic activity against SARS-CoV-2, ranging from 4.10- 5.93 μM/mL compared with Chloroquine as the standard drug with IC50 value about 2.24 μM/mL'' in line 153.
Author Response
Response to Reviewer 2
Comments and Suggestions for Authors
The authors have responded to all concerns that have been mentioned in the first report. However, there are still many major concerns that need to be addressed in their study:
the authors thank the reviewer for his benefit comments, and we appreciate your concern to publish our research work without any conflict, so all points were replayed as the following and any changes in the manuscript were highlighted with turquoise color.
Query (1): The way that the authors calculated/evaluated the activity of compounds using Plaque reduction assay is not logic at all. The equation that the authors followed does NOT give a real mathematical way for evaluating the activity. For example,
compd 3a (5uM) showed a reduction from 11x105 to 1.76x105 ~84% inhibition,
compound 4b (5uM) showed a reduction from 9x105 to 0.09x105 ~ 99% inhibition
Chloroquine 5uM- showed a reduction from 6x105 to ZERO ~100% inhibition
According to the presented calculation way, the difference between 99% inhibition and 100% inhibition is 0.09x105!!!!!! this is totally not logic!!!
Replay: Dear respect reviewer, the plaques were counted and the percentage reduction in plaques formation (% Reduction) in comparison to control wells was recorded according to the following Eq. (1) as following and provided in section 3.2.2. Plaque reduction assay:
(1)
Where, the viral count (untreated) is the viral count in wells where virus was untreated with the compounds. Additionally, the Viral count (treated) is the viral count in wells where virus was treated with the compounds. Additionally, the table 2 that represented the inhibition percentage now modified in the manuscript as following
Table 2. Inhibition % of the compounds 3a-c, 4a-e, 5a-e, and Chloroquine as reference drug
Cpd. No. |
Conc (µM) |
Viral count (untreated) (PFU/mL) |
Viral count (treated)(PFU/mL) |
Inhibition % |
3a |
5 |
11 X 105 |
1.76 X 105 |
84 |
2.5 |
1.98 X 105 |
82 |
||
1.25 |
3.3 X 105 |
70 |
||
0.625 |
3.85 X 105 |
65 |
||
3b |
5 |
10 X 105 |
2.5 X 105 |
75 |
2.5 |
4.3 X 105 |
57 |
||
1.25 |
5 X 105 |
50 |
||
0.625 |
8.3 X 105 |
17 |
||
3c |
5 |
5 X 105 |
5 X 105 |
0 |
2.5 |
5 X 105 |
0 |
||
1.25 |
5 X 105 |
0 |
||
0.625 |
5 X 105 |
0
|
||
3d |
5 |
11 X 105 |
4.95 X 105 |
55 |
2.5 |
6.6 X 105 |
40 |
||
1.25 |
7.7 X 105 |
30 |
||
0.625 |
9.9 X 105 |
10 |
||
4a |
5 |
10 X 105 |
10 X 105 |
0 |
2.5 |
10 X 105 |
0 |
||
1.25 |
10 X 105 |
0 |
||
0.625 |
10 X 105 |
0 |
||
4b |
5 |
9 X 105 |
0.09 X 105 |
99 |
2.5 |
0.9 X 105 |
90 |
||
1.25 |
1.8 X 105 |
80 |
||
0.625 |
2.7 X 105 |
70 |
||
4c |
5 |
11 X 105 |
2.86 X 105 |
74 |
2.5 |
3.85 X 105 |
65 |
||
1.25 |
4.4 X 105 |
60 |
||
0.625 |
9.68 X 105 |
12 |
||
4d |
5 |
11 X 105 |
2.2 X 105 |
80 |
2.5 |
4.4 X 105 |
60 |
||
1.25 |
6.6 X 105 |
40 |
||
0.625 |
7.26 X 105 |
34 |
||
4e |
5 |
9 X 105 |
0.81 X 105 |
91 |
2.5 |
1.17 X 105 |
87 |
||
1.25 |
1.53 X 105 |
83 |
||
0.625 |
3.96 X 105 |
56 |
||
5a |
5 |
9 X 105 |
7.2 X 105 |
20 |
2.5 |
8.28 X 105 |
8 |
||
1.25 |
8.73 X 105 |
3 |
||
0.625 |
9 X 105 |
0 |
||
5b |
5 |
5 X 105 |
1.65 X 105 |
67 |
2.5 |
2.65 X 105 |
47 |
||
1.25 |
4.45 X 105 |
11 |
||
0.625 |
5 X 105 |
0 |
||
5c |
5 |
11 X 105 |
6.6 X 105 |
40 |
2.5 |
8.14 X 105 |
26 |
||
1.25 |
8.8 X 105 |
20 |
||
0.625 |
11 X 105 |
0 |
||
5d |
5 |
10 X 105 |
1.8 X 105 |
82 |
2.5 |
4.9 X 105 |
51 |
||
1.25 |
6 X 105 |
40 |
||
0.625 |
6 X 105 |
40 |
||
5e |
5 |
5 X 105 |
1.5 X 105 |
70 |
2.5 |
3 X 105 |
40 |
||
1.25 |
3.35 X 105 |
33 |
||
0.625 |
4.5 X 105 |
10 |
||
Chloroquine |
5 |
6X104 |
0 |
>99 |
2.5 |
0 |
>99 |
||
1.25 |
0 |
>99 |
||
0.625 |
0 |
>99 |
ــــــــــــــــــــــــــــــــــــــــــــــــــــــــــــــــــــــــــــــــــــــــــــــــــــــــــــــــــ
Query (2): The presented results from the plaque assay only based on ONLY one measurement which is doubting more about the real activity of the tested compounds.
Replay: Dear respect reviewer, the presented results from the plaque assay is average of three replicates for each measurement.
ــــــــــــــــــــــــــــــــــــــــــــــــــــــــــــــــــــــــــــــــــــــــــــــــــــــــــــــــــ
Query (2): It would be beneficial and informative if the authors included the IC50 graphs for the tested compounds to the supplementary information.
Replay: Dear respect reviewer, we now added the IC50 and Cytotoxicity assay graphs in supplementary material file. Besides, the IC50 and Cytotoxicity assay graphs provided in the revises as following
Anti-COVID Cpd report
- Cytotoxicity assay:
3A |
Conc (micro g/µl) |
Cytotoxicity% |
1000 |
74.9480 |
|
500 |
50.2564 |
|
250 |
28.876276 |
|
125 |
15.212 |
|
62.5 |
12.1323 |
|
31.2 |
5.35241 |
4A |
Conc (micro g/µl) |
Cytotoxicity% |
1000 |
75.9821 |
|
500 |
52.8768 |
|
250 |
29.78287 |
|
125 |
14.97829 |
|
62.5 |
11.87209 |
|
31.2 |
6.16757 |
5A |
Conc (micro g/µl) |
Cytotoxicity% |
1000 |
74.8721 |
|
500 |
50.13435 |
|
250 |
28.23546 |
|
125 |
16.8754 |
|
62.5 |
13.12375 |
|
31.2 |
5.83231 |
3C |
Conc (micro g/µl) |
Cytotoxicity% |
1000 |
76.7863 |
|
500 |
51.2423 |
|
250 |
29.2411 |
|
125 |
15.95432 |
|
62.5 |
11.6224 |
|
31.2 |
6.02111 |
4C |
Conc (micro g/µl) |
Cytotoxicity% |
1000 |
79.12321 |
|
500 |
41.45364 |
|
250 |
39.12891 |
|
125 |
28.11989 |
|
62.5 |
23.12398 |
|
31.2 |
15.02359 |
5C |
Conc (micro g/µl) |
Cytotoxicity% |
1000 |
85.9899 |
|
500 |
50.11032 |
|
250 |
37.12091 |
|
125 |
28.00918 |
|
62.5 |
12.97801 |
|
31.2 |
10.65442 |
3B |
Conc (micro g/µl) |
Cytotoxicity% |
1000 |
83.6251 |
|
500 |
53.8761 |
|
250 |
39.1991 |
|
125 |
22.0918 |
|
62.5 |
17.1098 |
|
31.2 |
4.65510 |
4B |
Conc (micro g/µl) |
Cytotoxicity% |
1000 |
77.0009 |
|
500 |
58.0981 |
|
250 |
49.1205 |
|
125 |
37.76181 |
|
62.5 |
25.1651 |
|
31.2 |
19.18054 |
5B |
Conc (micro g/µl) |
Cytotoxicity% |
1000 |
90.9826 |
|
500 |
60.2976 |
|
250 |
44.2976 |
|
125 |
30.0092 |
|
62.5 |
21.14378 |
|
31.2 |
16.1887 |
3D |
Conc (micro g/µl) |
Cytotoxicity% |
1000 |
89.5618 |
|
500 |
56.1097 |
|
250 |
44.2652 |
|
125 |
32.7651 |
|
62.5 |
21.1987 |
|
31.2 |
9.19871 |
4D |
Conc (micro g/µl) |
Cytotoxicity% |
1000 |
88.01981 |
|
500 |
55.66551 |
|
250 |
30.19711 |
|
125 |
23.19871 |
|
62.5 |
11.18711 |
|
31.2 |
2.187511 |
5D |
Conc (micro g/µl) |
Cytotoxicity% |
1000 |
73.14226 |
|
500 |
56.7811 |
|
250 |
29.0912 |
|
125 |
18.07912 |
|
62.5 |
14.7896 |
|
31.2 |
8.90569 |
4E |
Conc (micro g/µl) |
Cytotoxicity% |
1000 |
77.09444 |
|
500 |
50.9866 |
|
250 |
22.45123 |
|
125 |
14.3221 |
|
62.5 |
11.3308 |
|
31.2 |
2.30256 |
5E |
Conc (micro g/µl) |
Cytotoxicity% |
1000 |
88.1425 |
|
500 |
56.2314 |
|
250 |
22.2232 |
|
125 |
11.33456 |
|
62.5 |
10.98061 |
|
31.2 |
5.18705 |
ــــــــــــــــــــــــــــــــــــــــــــــــــــــــــــــــــــــــــــــــــــــــــــــــــــــــــــــــــ
4- Regarding the presented results from the docking study, the authors claimed that ''only the H-bonds can observe as well as the other interaction as Arene-arene interaction or Arene-Cation interaction appear only in 3D and not appear at the 2D view''. However, the 3D presented if Fig6b does NOT also show the Arene-Cation interaction mentioned in Table4?? this also applied for other examples, 4d (PDB 6vxx) in fig 6e. The authors should make shots for the 3D view which clearly shows the claimed interactions that they mentioned in the table.
Replay: Dear respect reviewer, we apologize for mistyping occurred in the previous revision about the answer for this query, where both arene-arene interaction or arene-cation interaction appear only in 2D and do not appear at the 3D view when the docking is performed using Molecular Operating Environmental (MOE) version 10.2008, and according to this version, only the H-bonds can observe. In the higher version starting from MOE 2009 the arene-cation interaction can only appear, but we don't have any version of this type of interaction (upgraded version). Furthermore, our research group published many articles, and the arene-cation interaction appears only in 2D structure. Besides, to confirm our replayed many papers published using MOE 10.2008 and the arene-cation and arene-arene interaction does not appear and don't belong to our lab and from different publishers as:
- Novel benzothiazole hybrids targeting EGFR: Design, synthesis, biological evaluation and molecular docking studies; Journal of Molecular Structure; Volume 1240, 15 September 2021, 130595 (arene-arene interaction not appear using MOE 10.2008).
- Design, Synthesis, Anticancer Evaluation and Molecular Modeling of Novel Estrogen Derivatives; Molecules 2019, 24(3), 416; https://doi.org/10.3390/molecules24030416. (Arene-cation interaction not appear in 3D structure using MOE10.2008).
- Design, Synthesis, Molecular Docking of Novel Substituted Pyrimidinone Derivatives as Anticancer Agents; POLYCYCLIC AROMATIC COMPOUNDS; https://doi.org/10.1080/10406638.2020.1837888 (Arene-cation interaction not appear 10.2008).
- Design, synthesis and molecular docking of new pyrazole-thiazolidinones as potent anti-inflammatory and analgesic agents with TNF-α inhibitory activity; Bioorganic Chemistry; Volume 111, June 2021, 104827; https://doi.org/10.1016/j.bioorg.2021.104827. (arene-cation and arene-arene interactions not appear in 3D figures using MOE 10.2008)
- Chiral Pyridine-3,5-bis- (L-phenylalaninyl-L-leucinyl) Schiff Base Peptides as Potential Anticancer Agents: Design, Synthesis, and Molecular Docking Studies Targeting Lactate Dehydrogenase-A; Molecules 2020, 25(5), 1096; https://doi.org/10.3390/molecules25051096 (arene-cation interaction not appear using MOE 10.2008).
- Design, Synthesis, Anticancer Evaluation, Enzymatic Assays, and a Molecular Modeling Study of Novel Pyrazole–Indole Hybrids; ACS Omega 2021, 6, 18, 12361–12374; https://doi.org/10.1021/acsomega.1c01604 (arene-cation interaction not appear MOE 10.2008)
Additionally, we changed the scale of the docking figures over the manuscript to become clearer for the reader.
ــــــــــــــــــــــــــــــــــــــــــــــــــــــــــــــــــــــــــــــــــــــــــــــــــــــــــــــــــ
Query (5): In my first report, I mentioned that the selection of the PDBs are not correct. Since there is no co-crystallized ligand, the applied protocol would not be applicable unless the authors have created a homology study for active site before docking. The authors have replied as ''Additionally, there are many different methods to generate the active site based on the program of docking and protocol used. our study based on using the dummy atoms to generate the active site and its this''
As I got the authors have performed their docking study by generating the active site by homology or other way. WHy these details are not mentioned in the manuscript? why the exact protocol that has been followed was not fully detailed in the material and methods section?
Replay: Dear respect reviewer, we added all details for docking simulation in the experimental section in the manuscript and now we highlighted this part. We don’t perform the homology modeling but the crystal structure of different proteins as pdb file for RNA-dependent RNA polymerase (RdRp) (PDB: 6M71) and spike glycoprotein (SGp) (PDB: 6VXX) were obtained from the protein data bank and that described in the manuscript. The first mention for (PDB: 6M71) in the manuscript [Gao, Y., Yan, L., Huang, Y., Liu, F., Zhao, Y., Cao, L., ... & Rao, Z. (2020). Structure of the RNA-dependent RNA polymerase from COVID-19 virus. Science, 368(6492), 779-782]. Besides, for spike glycoprotein (SGp) (PDB: 6VXX), it was published by [Walls, A. C., Park, Y. J., Tortorici, M. A., Wall, A., McGuire, A. T., & Veesler, D. (2020). Structure, function, and antigenicity of the SARS-CoV-2 spike glycoprotein. Cell, 181(2), 281-292]. Additionally, the structure of the newly designed derivatives were prepared by protonation, and the energy was minimized using MMFF94x forcefield.
For the active site: we reported that the active site were generated using dummy atoms after selected only one chains in both two proteins and that described in the manuscript as following: [The selected protein and docking process structure was prepared according to default protocol and according to the previously reported method [45] using the dummy atoms to generate the active site. We selected only one chain for SARS-CoV-2 spike glycoprotein (PDB: 6VXX) (Chain A). Also, the SARS-CoV-2 RNA-dependent RNA polymerase (PDB: 6M71) chain A was selected for docking proposed]. The previous method was used previously in the following manuscript (Fukuda, T.; Umeki, T.; Tokushima, K.; Xiang, G.; Yoshida, Y.; Ishibashi, F.; Oku, Y.; Nishiya, N.; Uehara, Y.; Iwao, M. Design, synthesis, and evaluation of A-ring-modified lamellarin N analogues as noncovalent inhibitors of the EGFR T790M/L858R mutant. Bioorg. Med. Chem. 2017, 25, 6563–6580, doi:https://doi.org/10.1016/j.bmc.2017.10.030). Additionally, in the dummy atom methods we selected the higher sequence from the amino acids.
For docking process: we reported in the manuscript both placement method and scoring function and the following paragraph present in the experimental section of the manuscript and now highlighted as [The docking simulation inside the active site was performed using the Trigonal matcher placement method using London dG as a scoring function].
For energy obtained from docking process that revealed to strength of binding we mention in the manuscript that following sentence (Its energy represented by Kcal/mol and the top-scoring pose were inspected visually).
All the above details about docking simulation process and protocol now highlighted in the manuscript. Additionally, we appreciate your concern, but many reported papers with different publishers used the same PDB files as
Papers published with of RNA-dependent RNA polymerase (RdRp) (PDB: 6M71):
- Esam, Z., Akhavan, M., & Bekhradnia, A. (2022). Molecular docking and dynamics studies of Nicotinamide Riboside as a potential multi-target nutraceutical against SARS-CoV-2 entry, replication, and transcription: A new insight. Journal of molecular structure, 1247, 131394.
- Sarma, Himakshi, Esther Jamir, and G. Narahari Sastry. "Protein-protein interaction of RdRp with its co-factor NSP8 and NSP7 to decipher the interface hotspot residues for drug targeting: A comparison between SARS-CoV-2 and SARS-CoV." Journal of Molecular Structure (2022): 132602.
- Raj, Vinit, et al. "Antiviral activities of 4H-chromen-4-one scaffold-containing flavonoids against SARS–CoV–2 using computational and in vitro approaches." Journal of Molecular Liquids (2022): 118775.
- Gao, Yan, et al. "Structure of the RNA-dependent RNA polymerase from COVID-19 virus." Science 368.6492 (2020): 779-782.
- Elkarhat, Z., Charoute, H., Elkhattabi, L., Barakat, A., & Rouba, H. (2022). Potential inhibitors of SARS-cov-2 RNA dependent RNA polymerase protein: molecular docking, molecular dynamics simulations and MM-PBSA analyses. Journal of Biomolecular Structure and Dynamics, 40(1), 361-374.
- Brogi, S., Quimque, M. T., Notarte, K. I., Africa, J. G., Hernandez, J. B., Tan, S. M., ... & Macabeo, A. P. (2022). Virtual Combinatorial Library Screening of Quinadoline B Derivatives against SARS-CoV-2 RNA-Dependent RNA Polymerase. Computation, 10(1), 7.
- Elfiky, A. A., Mahran, H. A., Ibrahim, I. M., Ibrahim, M. N., & Elshemey, W. M. (2022). Molecular dynamics simulations and MM-GBSA reveal novel guanosine derivatives against SARS-CoV-2 RNA dependent RNA polymerase. RSC Advances, 12(5), 2741-2750.
and so on
Paper published with spike glycoprotein (SGp) (PDB: 6VXX):
- Al Khoury, C., Bashir, Z., Tokajian, S., Nemer, N., Merhi, G., & Nemer, G. (2022). In silico evidence of beauvericin antiviral activity against SARS-CoV-2. Computers in biology and medicine, 141, 105171.
- Rashid, H., Ahmad, N., Abdalla, M., Khan, K., Martines, M. A. U., & Shabana, S. (2022). Molecular docking and dynamic simulations of Cefixime, Etoposide and Nebrodenside A against the pathogenic proteins of SARS-CoV-2. Journal of Molecular Structure, 1247, 131296.
- Dawood, A. A. (2022). Increasing the frequency of omicron variant mutations boosts the immune response and may reduce the virus virulence. Microbial Pathogenesis, 105400.
- Dhameliya, T. M., Nagar, P. R., & Gajjar, N. D. (2022). Systematic virtual screening in search of SARS CoV-2 inhibitors against spike glycoprotein: pharmacophore screening, molecular docking, ADMET analysis and MD simulations. Molecular diversity, 1-18.
and so on.
ــــــــــــــــــــــــــــــــــــــــــــــــــــــــــــــــــــــــــــــــــــــــــــــــــــــــــــــــــ
Query (6): lastly, the authors mentioned in their rebuttal ''this work was done through collaboration with the Main Biological Warfare Laboratories, Chemical Warfare Department, Ministry of Defense, Cairo, Egypt. This was through a protocol of cooperation between the Faculty of Pharmacy (Girls)/ Faculty of Science, Al-Azhar Unveracity, and the Ministry of Defense (If you need a copy, I can send it)''
I just was wondering if this project was mainly collaboration Al-Azhar Unveracity, and the Ministry of Defense in Egypt. What has been done in Taif University in Saudi Arabia ??
Replay: Dear respect reviewer, this answer for the query asked in the previous revise that (there is NO biomolecular study that can PROVE that the authors have performed any kind of biological assays to confirm the activity of compounds!!) and we answer that the biological parts was performed through (collaboration with the Main Biological Warfare Laboratories, Chemical Warfare Department, Ministry of Defense, Cairo, Egypt. This was through a protocol of cooperation between the Faculty of Pharmacy (Girls)/ Faculty of Science, Al-Azhar Unveracity, and the Ministry of Defense (If you need a copy, I can send it). Additionally, the main backbone part for our work (chemistry design and synthesis of the newly designed compounds) that carried out mainly in the Faculty of Science at Al-Azhar University (staff from the chemistry department) performed with the help of two other departments (Chemistry and pharmaceutical organic chemistry departments) as described in the author contribution as following:
- Department of Chemistry, College of Science, Taif University for Dr Ola A. Abu Ali.
- Pharmaceutical Organic Chemistry Department, Faculty of Pharmacy (Girls), Al-Azhar University for Dr. Samar A. El‑Kalyoubi and Dr. Eman A. Fayed.
these three authors that mentioned previously didn’t late for any help from chemicals used in the design of the synthesized derivatives and interpret the data with remove any conflict and work side by side with two authors belong to the chemistry department in the Faculty of Science- Al-Azhar University. Additionally, the authors contribution was provided in the manuscript.
ــــــــــــــــــــــــــــــــــــــــــــــــــــــــــــــــــــــــــــــــــــــــــــــــــــــــــــــــــ
Minor issues;
- Query (1): The authors should notice that there is no unite identified as uM/mL yet. Please correct this through the manuscript.
Replay: Dear respect reviewer, we modified them and highlighted the modified part in the manuscript through the section 2.1.1. The half-maximal inhibitory concentration (IC50).
ــــــــــــــــــــــــــــــــــــــــــــــــــــــــــــــــــــــــــــــــــــــــــــــــــــــــــــــــــ
- Query (2): the cytotoxicity evaluation was NOT performed for SARsCov-2 -infected cells. Accordingly, the authors can NOT say that ''The synthetic compounds 3a-d, 4a-e, and 5a-e were tested in vitro against the SARS-CoV-2 strain in Egyptian patients'' in line 148 or ''According to the results, most of the compounds tested showed moderate to excellent cytotoxic activity against SARS-CoV-2, ranging from 4.10- 5.93 μM/mL compared with Chloroquine as the standard drug with IC50 value about 2.24 μM/mL'' in line 153.
Replay: Dear respect reviewer, we modified this sentence and highlighted in the manuscript as (The synthetic compounds 3a-d, 4a-e, and 5a-e were tested in vitro against the SARS-CoV-2 strain isolated from Egyptian patients). Where our work performed on strain isolated from Egyptian patients that recorded on the GenBank with code (MT776904) by Egy-Army-MCL001/2020 and the link of this on site represented as (https://www.ncbi.nlm.nih.gov/nuccore/MT776904) under the title :( Severe acute respiratory syndrome coronavirus 2 isolate SARS-CoV-2/human/EGY/Egy-Army-MCL001/2020, complete genome).
Reviewer 4 Report
Authors did the suggested corrections and thus the manuscript is suitable for publication in Pharmaceuticals.
Author Response
many thanks